# Understanding Square Loss in Training Overparameterized Neural Network Classifiers

**Tianyang Hu**[*]
Huawei Noah's Ark Lab
hutianyang1@huawei.com

**Jun Wang**[*]
HKUST
jwangfx@connect.ust.hk

**Wenjia Wang**[*]
HKUST (GZ) and HKUST
wenjiawang@ust.hk

**Zhenguo Li**
Huawei Noah's Ark Lab
li.zhenguo@huawei.com

## Abstract

Deep learning has achieved many breakthroughs in modern classification tasks. Numerous architectures have been proposed for different data structures but when it comes to the loss function, the cross-entropy loss is the predominant choice. Recently, several alternative losses have seen revived interests for deep classifiers. In particular, empirical evidence seems to promote square loss but a theoretical justification is still lacking. In this work, we contribute to the theoretical understanding of square loss in classification by systematically investigating how it performs for overparameterized neural networks in the neural tangent kernel (NTK) regime. Interesting properties regarding the generalization error, robustness, and calibration error are revealed. We consider two cases, according to whether classes are separable or not. In the general non-separable case, fast convergence rate is established for both misclassification rate and calibration error. When classes are separable, the misclassification rate improves to be exponentially fast. Further, the resulting margin is proven to be lower bounded away from zero, providing theoretical guarantees for robustness. We expect our findings to hold beyond the NTK regime and translate to practical settings. To this end, we conduct extensive empirical studies on practical neural networks, demonstrating the effectiveness of square loss in both synthetic low-dimensional data and real image data. Comparing to cross-entropy, square loss has comparable generalization error but noticeable advantages in robustness and model calibration. Code will be available at https://gitee.com/mindspore/models/tree/master/research/cv/sl_classification.

## 1 Introduction

The pursuit of better classifiers has fueled the progress of machine learning and deep learning research. The abundance of benchmark image datasets, e.g., MNIST, CIFAR, ImageNet, etc., provide test fields for all kinds of new classification models, especially those based on deep neural networks (DNN). With the introduction of CNN, ResNets, and transformers, DNN classifiers are constantly improving and catching up to the human-level performance. In contrast to the active innovations in model architecture, the training objective remains largely stagnant, with cross-entropy loss being the default choice. Despite its popularity, cross-entropy has been shown to be problematic in some applications. Among others, [1] argued that features learned from cross-entropy lack interpretability and proposed a new loss aiming for maximum coding rate reduction. [2] linked the use of cross-entropy to adversarial

---

[*]These authors contributed equally to this work. Corresponding author: Wenjia Wang.

36th Conference on Neural Information Processing Systems (NeurIPS 2022).

vulnerability and proposed a new classification loss based on latent space matching. [3] discovered that the confidence of most DNN classifiers trained with cross-entropy is not well-calibrated.

Recently, several alternative losses have seen revived interests for deep classifiers. In particular, many existing works have presented empirical evidence promoting the use of square loss over cross-entropy. [4] conducted large-scale experiments comparing the two and found that square loss tends to perform better in natural language processing related tasks while cross-entropy usually yields slightly better accuracy in image classification. Similar comparisons are also made in [5, 6, 7, 8]. [9] compared a variety of loss functions and output layer regularization strategies on the accuracy and out-of-distribution robustness, and found that square loss has greater class separation and better performance.

In comparison to the empirical investigation, theoretical understanding of square loss in training deep learning classifiers is still lacking. Through our lens, square loss has its uniqueness among classic classification losses, and we argue that it has great potentials for modern classification tasks. Below we list our motivations and reasons why.

**Explicit feature modeling**    Deep learning's success can be largely attributed to its superior ability as feature extractors. For classification, the ideal features should be separated between classes and concentrated within classes. However, when optimizing cross-entropy loss, it is not obvious what the learned features should look like [1]. In the terminal stage of training, [10] proved that when cross-entropy is sufficiently minimized, the penultimate layers features will collapse to the scaled simplex structure. Such phenomenon is referred to as "neural collapse". Knowing this, would it be better to directly enforce the terminal solution by using square loss with the simplex coding [11], as Euclidean distance is probably most natural to measure the distance between samples' features and class-means? Unlike cross-entropy, square loss uses the label codings (one-hot, simplex etc.) as features, which can explicitly control class separations.

**Model calibration**    An ideal classifier should not only give the correct class prediction, but also with the correct confidence. Calibration error measures the closeness of the predicted confidence to the underlying conditional probability $\eta$. Using square loss in classification can be essentially viewed as regression where it treats discrete labels as continuous code vectors. It can be shown that the optimal classifier under square loss is $2\eta - 1$, linear with the ground truth. This distinguishing property allows it to easily recover $\eta$. In comparison, the optimal classifiers under the hinge loss and cross-entropy are $\text{sign}(2\eta - 1)$ and $\log(\frac{\eta}{1-\eta})$, respectively [12]. Therefore, hinge loss doesn't provide reliable information on the prediction confidence, and cross-entropy can be problematic when $\eta$ is close to 0 or 1, as stated in [12]. Hence, in terms of model calibration, square loss is a natural choice.

**Connections to popular approaches**    Mixup [13] is a data augmentation technique where augmented data are constructed via convex combinations of inputs and their labels. Like in square loss, mixup treats labels as continuous and is shown to improve the generalization of DNN classifiers. In knowledge distillation [14], where a student classifier is trying to learn from a trained teacher, [15] proved that the "best" teacher with the ground truth conditional probabilities provides the lowest variance in student learning. Since classifiers trained using square loss is a consistent estimator of $\eta$, one can argue that it is a better teacher. In supervised contrastive learning [16], the optimal features are the same as those from square loss with simplex label coding [17] (details in Section 3.4).

Despite its lack of popularity in practice, square loss has many advantages that can be easily overlooked. In this work, we systematically investigate from a statistical estimation perspective, the properties of deep learning classifiers trained using square loss. Comparing to cross entropy, the square loss is much more theoretically tractable, which allows us to develop sharper results on the training process and generalization performance. The neural networks in our analysis are required to be sufficiently overparameterized in the neural tangent kernel (NTK) regime. Even though this restricts the implication of our results, it is a necessary first step towards a deeper understanding. In summary, our main contributions are:

- **Generalization error bound**: We consider two cases, according to whether classes are separable or not. In the general non-separable case, we adopt the classical binary classification setting with smooth conditional probability. Fast convergence rate is established for overparameterized neural network classifiers with Tsybakov's noise condition. If two

classes are separable with positive margin, we show that overparameterized neural network classifiers can provably reach zero misclassification error with probability *exponentially* tending to one. To the best of our knowledge, this is the *first* such result for separable but not linear separable classes. Furthermore, we bridge these two cases and offer a *unified* view by considering auxiliary random noise injection.

- **Adversarial robustness (margin property)**: When two classes are separable, the decision boundary is not unique and large-margin classifiers are preferred. The margin is naturally connected to adversarial robustness [18]. In the separable case, we show that the decision boundary of overparameterized neural network classifiers trained by square loss cannot be too close to the data support and the resulting margin is lower bounded away from zero, providing theoretical guarantees for adversarial robustness.

- **Calibration error**: We show that classifiers trained using square loss are inherently well-calibrated, i.e., the trained classifier provides consistent estimation of the ground-truth conditional probability in $L_\infty$ norm. Such property doesn't hold for cross-entropy.

- **Empirical evaluation**: We corroborate our theoretical findings with empirical experiments in both synthetic low-dimensional data and real image data. Comparing to cross-entropy, square loss has comparable generalization error but noticeable advantages in robustness and model calibration.

This work contributes to the theoretical understanding of deep classifiers from a nonparametric estimation point of view, which has been a classic topic in statistics literature. Among others, [19] established the optimal convergence rate for 0-1 loss excess risk when the decision boundary is smooth. [12, 20] extended the analysis to various surrogate losses. [21, 22] studied the convergence rates for plug-in classifiers from local averaging estimators. [23] investigated the convergence rate for support vector machine using Gaussian kernels. We build on and extend classic results to neural networks in the NTK regime. There exist nonparametric results on deep classifiers, e.g., fast convergence rates have been derived for DNN classifiers that minimize the empirical 0-1 loss [24, 25], hinge loss [26] and cross-entropy loss [27], etc. Unlike aforementioned works that only concern the existence of a good classifier (with theoretical worst-case guarantee), in ignorance of the tremendous difficulty of neural network optimization, our results further incorporate the training algorithm and apply to trained classifiers, which relates better to practice. To the best of the authors' knowledge, similar attainable fast rates (faster than $n^{-\frac{1}{2}}$) have never been established for neural network classifiers.

We require the neural network to be overparameterized, which has been extensively studied recently via NTK. Most such results are in the regression setting with a handful of exceptions. [28] showed that only polylogarithmic width is sufficient for gradient descent to overfit the training data using logistic loss. [29] proved generalization error bound for regularized NTK in classification. [30, 31] provided optimization and generalization guarantees for overparameterized network trained with cross-entropy. In comparison, our results are sharper in the sense that we take the ground truth data assumptions into consideration. This allows a faster convergence rate, especially when the classes are separable, where the exponential convergence rate is attainable. The NTK framework greatly reduces the technical difficulty for our theoretical analysis. However, our results are mainly due to properties of the square loss itself and we expect them to hold for a wide range of classifiers.

There are other works investigating the use of square loss for training (deep) classifiers. [32] uncovered that the "neural collapse" phenomenon also occurs under the square loss. [33] compared classification and regression tasks in the overparameterized linear model with Gaussian features, illustrating different roles and properties of loss functions used at the training and testing phases. [34] made interesting observations on effects of popular regularization techniques such as batch normalization and weight decay on the gradient flow dynamics under square loss. These findings support our theoretical results' implication, which further strengthens our beliefs that the essence comes from the square loss and our analysis can go beyond NTK regime.

The rest of this paper is arranged as follows. Section 2 presents some preliminaries. Main theoretical results are in Section 3. The simplex label coding is discussed in Section 3.4 followed by numerical studies in Section 4 and conclusions in Section 5. Technical proofs and details of the numerical studies can be found in the Appendix.

## 2 Preliminaries

**Notation** For a function $f : \Omega \to \mathbb{R}$, let $\|f\|_\infty = \sup_{\boldsymbol{x} \in \Omega} |f(\boldsymbol{x})|$ and $\|f\|_p = (\int_\Omega |f(\boldsymbol{x})|^p d\boldsymbol{x})^{1/p}$. For a vector $\boldsymbol{x}$, $\|\boldsymbol{x}\|_p$ denotes its $p$-norm, for $1 \le p \le \infty$. $L_p$ and $l_p$ are used to distinguish function norms and vector norms. For two positive sequences $\{a_n\}_{n \in \mathbb{N}}$ and $\{b_n\}_{n \in \mathbb{N}}$, we write $a_n \lesssim b_n$ if there exists a constant $C > 0$ such that $a_n \le C b_n$ for all sufficiently large $n$. We write $a_n \asymp b_n$ if $a_n \lesssim b_n$ and $b_n \lesssim a_n$. Let $[N] = \{1, \ldots, N\}$ for $N \in \mathbb{N}$, $\mathbb{I}$ be the indicator function, and $\boldsymbol{I}_d$ be the $d \times d$ identity matrix. $N(\mu, \boldsymbol{\Sigma})$ represents Gaussian distribution with mean $\mu$ and covariance $\boldsymbol{\Sigma}$.

**Classification problem settings** Let $P$ be an underlying probability measure on $\Omega \times \boldsymbol{Y}$, where $\Omega \subset \mathbb{R}^d$ is compact and $\boldsymbol{Y} = \{1, -1\}$. Let $(X, Y)$ be a random variable with respect to $P$. Suppose we have observations $\{(\boldsymbol{x}_i, y_i)\}_{i=1}^n \subset (\Omega \times Y)^n$ i.i.d. sampled according to $P$. The classification task is to predict the unobserved label $y$ given a new input $\boldsymbol{x} \in \Omega$. Let $\eta$ defined on $\Omega$ denote the conditional probability, i.e., $\eta(\boldsymbol{x}) = \mathbb{P}(y = 1|\boldsymbol{x})$. Let $P_X$ be the marginal distribution of $P$ on $X$.

**Remark 1.** In this work, we consider the fixed dimension settings, i.e., $d$ is fixed. As pointed by a reviewer, [35, 36] discussed overparameterized neural networks in high dimensional settings under regression. The study in the high dimensional settings will be pursued in the future.

According to whether labels are deterministic, there are two scenarios of interest. If $\eta$ only takes values from $\{0, 1\}$, i.e., labels are deterministic, we call this case the *separable case*. Note that in the separable case we consider, the classes are not limited to linearly separable but can be arbitrarily complicated, as long as the label is deterministic. Let $\Omega_1 = \{\boldsymbol{x}|\eta(\boldsymbol{x}) = 1\}$, $\Omega_2 = \{\boldsymbol{x}|\eta(\boldsymbol{x}) = 0\}$ and $\Omega = \Omega_1 \cup \Omega_2$. If the probability measure of $\{\boldsymbol{x}|\eta(\boldsymbol{x}) \in (0, 1)\}$ is non-zero, i.e., the labels contain randomness, we call this case the *non-separable case*. In the separable case, we further assume that there exists a positive margin, i.e., $\text{dist}(\Omega_1, \Omega_2) \ge 2\gamma > 0$, where $\gamma$ is a constant, and $\text{dist}(\Omega_1, \Omega_2) = \inf_{\boldsymbol{x} \in \Omega_1, \boldsymbol{x}' \in \Omega_2} \|\boldsymbol{x} - \boldsymbol{x}'\|_2$. In the non-separable case, to quantify the difficulty of classification, we adopt the well-established Tsybakov's noise condition [21], which measures how large the "difficult region" is where $\eta(\boldsymbol{x}) \approx 1/2$.

**Definition 2.1** (Tsybakov's noise condition). Let $\kappa \in [0, \infty]$. We say $P$ has Tsybakov noise exponent $\kappa$ if there exists a constant $C, T > 0$ such that for all $0 < t < T$, $P_X(|2\eta(X) - 1| < t) \le C \cdot t^\kappa$.

A large value of $\kappa$ implies the difficult region to be small. It is expected that a larger $\kappa$ leads to a faster convergence rate of a neural network classifier. This intuition is verified for the overparameterized neural network classifier trained by square loss and $\ell_2$ regularization. See Section 3 for details.

The key quantity of interest is the misclassification error, i.e., 0-1 loss. In the population level, the 0-1 loss can be written as

$$
\begin{aligned}
L(f) &= \mathbb{E}_{(X,Y) \sim P} \mathbb{I}\{\text{sign}(f(X)) \ne Y\} \\
&= \mathbb{E}_{X \sim P_X}[(1 - \eta(X))\mathbb{I}\{f(X) \ge 0\} + \eta(X)\mathbb{I}\{f(X) < 0\}].
\end{aligned} \tag{2.1}
$$

One optimal minimizer of $L(f)$ is $2\eta - 1$, and we define $L^* = L(2\eta - 1)$. In fact, as long as the classifier has the same sign with $2\eta(\boldsymbol{x}) - 1$ for all $\boldsymbol{x} \in \Omega$, it is a optimal minimizer and achieves the minimum 0-1 loss.

**Neural network setup** We mainly focus on the one-hidden-layer ReLU neural network family $\mathcal{F}$ with $m$ nodes in the hidden layer, denoted by $f_{\boldsymbol{W}, \boldsymbol{a}}(\boldsymbol{x}) = m^{-1/2} \sum_{r=1}^m a_r \sigma(\boldsymbol{W}_r^\top \boldsymbol{x})$, where $\boldsymbol{x} \in \Omega$, $\boldsymbol{W} = (\boldsymbol{W}_1, \cdots, \boldsymbol{W}_m) \in \mathbb{R}^{d \times m}$ is the weight matrix in the hidden layer, $\boldsymbol{a} = (a_1, \cdots, a_m)^\top \in \mathbb{R}^m$ is the weight vector in the output layer, $\sigma(z) = \max\{0, z\}$ is the rectified linear unit (ReLU). The initial values of the weights are independently generated from $\boldsymbol{W}_r(0) \sim N(\boldsymbol{0}, \xi^2 \boldsymbol{I}_m), a_r \sim \text{unif}\{-1, 1\}, \forall r \in [m]$. Based on the observations $\{(\boldsymbol{x}_i, y_i)\}_{i=1}^n$, the goal of training a neural network is to find a solution to

$$
\min_{\boldsymbol{W}} \sum_{i=1}^n l(f_{\boldsymbol{W}, \boldsymbol{a}}(\boldsymbol{x}_i), y_i) + \mu \mathcal{R}(\boldsymbol{W}, \boldsymbol{a}), \tag{2.2}
$$

where $l$ is the loss function, $\mathcal{R}$ is the regularization, and $\mu \ge 0$ is the regularization parameter. Note in Equation 2.2 that we only consider training the weights $\boldsymbol{W}$. This is because $a \cdot \sigma(z) = \text{sign}(a) \cdot \sigma(|a|z)$,

which allows us to reparametrize the network to have all $a_i$'s to be either 1 or $-1$. In this work, we consider square loss associated with $\ell_2$ regularization, i.e., $l(f_{\boldsymbol{W},\boldsymbol{a}}(\boldsymbol{x}_i), y_i) = (f_{\boldsymbol{W},\boldsymbol{a}}(\boldsymbol{x}_i) - y_i)^2$ and $\mathcal{R}(\boldsymbol{W},\boldsymbol{a}) = \|\boldsymbol{W}\|_2^2$.

A popular way to train the neural network is via gradient based methods. It has been shown that the training process of DNNs can be characterized by the neural tangent kernel (NTK) [37]. As is usually assumed in the NTK literature [38, 29, 39, 40], we consider data on the unit sphere $\mathbb{S}^{d-1}$, i.e., $\|\boldsymbol{x}_i\|_2 = 1, \forall i \in [n]$, and the neural network is highly overparameterized ($m \gg n$) and trained by gradient descent (GD). For details about NTK and GD in one-hidden-layer ReLU neural networks, we refer to Appendix A. In the rest of this work, $f_{\boldsymbol{W}(k),\boldsymbol{a}}$ denotes the GD-trained neural network classifier under square loss associated with $\ell_2$ regularization, where $k$ is the iteration number satisfying Assumption D.1 and $\boldsymbol{W}(k)$ is the weight matrix after $k$-th iteration.

# 3 Theoretical results

In this section, we present our main theoretical results, which consist of three parts: generalization error, robustness, and calibration error. Throughout the analysis, we make the following assumptions on the data and the estimation model. Due to the page limit, we move the technical specification of the assumptions to Appendix D, with detailed discussions.

**Data assumptions** We assume the ground-truth $\eta(\boldsymbol{x})$ to be well-behaved (Assumption D.2). Optionally, the marginal density $X$ is assumed to be upper bounded (Assumptions D.4) or both upper and lower bounded (Assumptions D.5). Assumptions D.4 and D.5 are standard in classical analysis of classification in statistics literature [21, 22], which covers a large class of density functions.

**Model assumptions** We require the ReLU neural network to be sufficiently overparameterized (with a finite width), and imposes conditions on the learning rate and iteration number (Assumption D.1); similar settings have been adopted by [38, 40]. We also assume that the solution to (2.2) is well-behaved, i.e., the complexity of the neural network estimator generated by the GD training is controlled (Assumption D.3).

## 3.1 Generalization error bound

In classification, the generalization error is typically referred to as the misclassification error, which can be quantified by $L(f)$ defined in Equation 2.1. In the non-separable case, the excess risk, defined by $L(f) - L^*$, is used to evaluate the quality of a classifier $f$, where $L^* = L(2\eta - 1)$, which is the minimum 0-1 loss, as stated after Equation 2.1. Theorem 3.1 states that the overparameterized neural network with GD and $\ell_2$ regularization can achieve a small excess risk in the non-separable case.

**Theorem 3.1** (Excess risk in the non-separable case). Suppose Assumptions D.1, D.2, and D.4 hold. Assume the conditional probability $\eta(\boldsymbol{x})$ satisfies Tsybakov's noise condition with component $\kappa$. Let $\mu \asymp n^{\frac{d-1}{2d-1}}$. Then

$$L(f_{\boldsymbol{W}(k),\boldsymbol{a}}) = L^* + O_{\mathbb{P}}(n^{-\frac{d(\kappa+1)}{(2d-1)(\kappa+2)}}). \tag{3.1}$$

From Theorem 3.1, we can see that as $\kappa$ becomes larger, the convergence rate becomes faster, which is intuitively true. Generalization error bounds in this setting is scarce. To the best of the authors' knowledge, [29] is the closest work (the labels are randomly flipped), where the bound is in the order of $O_{\mathbb{P}}(1/\sqrt{n})$. Our bound is faster, especially with larger $\kappa$. It is known that the optimal convergence rate under Assumptions D.2 (with slightly different notion of the smoothness) and D.4 is $O_{\mathbb{P}}(n^{-\frac{d(\kappa+1)}{d\kappa+4d-2}})$ [21]. The differences between (3.1) and the optimal convergence rate is the extra $(d-1)\kappa$ in the denominator of the convergence rate in (3.1) (since $n^{-\frac{d(\kappa+1)}{(2d-1)(\kappa+2)}} = n^{-\frac{d(\kappa+1)}{(d-1)\kappa+d\kappa+4d-2}}$). If the conditional probability $\eta$ has a bounded Lipschitz constant, [22] showed that the convergence rate based on the plug-in kernel estimate is $O_{\mathbb{P}}(n^{-\frac{\kappa+1}{\kappa+3+d}})$, which is slower than the rate in Equation 3.1 if $d$ is large.

Now we turn to the separable case. Since $\eta$ only takes value from $\{0, 1\}$ in the separable case, $\eta$ is bounded away from $1/2$. Therefore, one can trivially take $\kappa \to \infty$ in Equation 3.1 and obtain the

convergence rate $O_{\mathbb{P}}(n^{-d/(2d-1)})$. However, this rate can be significantly improved in the separable case, as stated in the following theorem.

**Theorem 3.2** (Generalization error in the separable case). Suppose Assumptions D.1, D.3, and D.5 hold. Let $\mu = o(1)$. There exist positive constants $C_1, C_2$ such that the misclassification rate is 0% with probability at least $1 - \delta - C_1 \exp(-C_2 n)$, and $\delta$ can be arbitrarily small[2] by enlarging the neural network's width.

In Theorem 3.2, the regularization parameter can take any rate that converges to zero. In particular, $\mu$ can be zero, and the corresponding classifier overfits the training data. Theorem 3.2 states that the convergence rate in the separable case is exponential, if a sufficiently wide neural network is applied. This is because the observed labels are not corrupted by noise, i.e., $\mathbb{P}(y = 1|\boldsymbol{x})$ is either one or zero. Therefore, it is easier to classify separable data, which is intuitively true.

## 3.2 Robustness and calibration error

If two classes are separable with positive margin, the decision boundary is not unique. Practitioners often prefer the decision boundary with large margins, which are robust against possible perturbation on input points [41, 42]. The following theorem states that the square loss trained margin can be lower bounded by a positive constant. Recall that in the separable case, $\Omega = \Omega_1 \cup \Omega_2$, where $\Omega_1 = \{\boldsymbol{x}|\eta(\boldsymbol{x}) = 1\}$ and $\Omega_2 = \{\boldsymbol{x}|\eta(\boldsymbol{x}) = 0\}$.

**Theorem 3.3** (Robustness in the separable case). Suppose the assumptions of Theorem 3.2 are satisfied. Let $\mu = o(1)$. Then there exist positive constants $C, C_1, C_2$ such that $\min_{\boldsymbol{x} \in \mathcal{D}_T, \boldsymbol{x}' \in \Omega_1 \cup \Omega_2} \|\boldsymbol{x} - \boldsymbol{x}'\|_2 \geq C$, and the misclassification rate is 0% with probability at least $1 - \delta - C_1 \exp(-C_2 n)$ for all $n$, where $\mathcal{D}_T$ is the decision boundary, and $\delta$ is as in Theorem 3.2.

Theorem 3.3 states that the square loss trained margin is robust in the sense that the predicted label will not change in case of any noise whose $l_2$ norm is smaller than C. Since $\|\boldsymbol{x} - \boldsymbol{x}'\|_\infty \geq \sqrt{d} \|\boldsymbol{x} - \boldsymbol{x}'\|_2$, Theorem 3.3 also indicates $l_\infty$ robustness. To the best of our knowledge, similar theoretical robustness guarantee has not been provided for any other loss functions. The most relevant work is [43], but their result is not on the population and doesn't apply to ReLU networks trained via GD.

In the non-separable case, $\eta(\boldsymbol{x})$ varies within (0,1) and practitioners may not only want a classifier with a small excess risk, but also want to recover the underlying conditional probability $\eta$. Therefore, square loss is naturally preferred since it treats the classification problem as a regression problem. The following theorem states that, one can recover the conditional probability $\eta$ by using an overparameterized neural network with $\ell_2$ regularization and GD training.

**Theorem 3.4** (Calibration error). Suppose Assumptions D.1-D.4 are fulfilled. Let $\mu \asymp n^{\frac{d-1}{2d-1}}$. Then

$$\left\|(f_{\boldsymbol{W}(k),\boldsymbol{a}} + 1)/2 - \eta\right\|_{L_\infty} = O_{\mathbb{P}}(n^{-1/(4d-2)}).$$

Theorem 3.4 states that the underlying conditional probability in the non-separable case can be recovered by $(f_{\boldsymbol{W}(k),\boldsymbol{a}} + 1)/2$. The form $(f_{\boldsymbol{W}(k),\boldsymbol{a}} + 1)/2$ is to account for the $\{-1, 1\}$ label coding. Under $\{0,1\}$ coding, the estimator would be $f_{\boldsymbol{W}(k),\boldsymbol{a}}$ itself. The $L_\infty$ consistency doesn't hold for cross-entropy trained neural networks, due to the form of the optimal solution $\log(\frac{\eta}{1-\eta})$ [12]. With limited capacity, the network's confidence prediction is bounded away from 0 and 1. In practice, we want to control the complexity of the neural network thus it is usually the case that $\left\|f_{\boldsymbol{W}(k),\boldsymbol{a}}\right\|_\infty < C$ for some constant $C$. Hence, following the results in [12], direct calculation shows that such a neural network with limited complexity cannot accurately estimate $\eta(\boldsymbol{x})$ when $\eta(\boldsymbol{x}) > \frac{e^C}{1+e^C}$ or $\eta(\boldsymbol{x}) < \frac{1}{1+e^C}$, which makes the calibration error under the cross-entropy loss always bounded away from zero. However, square loss does not have such a problem.

Notice that the calibration error bound in Theorem 3.4 does not depend on the Tsybakov's noise condition, and is slower than the excess risk. This is because a small calibration error is much stronger than a small excess risk, since the former requires the conditional probability estimation to

---

[2]The term $\delta$ only depends on the width of the neural network. A smaller $\delta$ requires a wider neural network. If $\delta = 0$, then the number of nodes in the hidden layer is infinity.

be *uniformly* accurate, not just matching the sign of $\eta - 1/2$. To be more specific, a good estimated $\widehat{\eta}$ can always lead to a low risk plug-in classifier $\widehat{f} = 2\widehat{\eta} - 1$, but not vice versa.

**Remark 2** (Technical challenge). Despite the similar forms of regression and classification using square loss, most of the regression analysis techniques cannot be directly applied to the classification problem, even if the supports of two classes are non-separable. Moreover, it is clear that classification problems in the separable case are completely different with regression problems.

### 3.3 Transition from separable to non-separable

The general non-separable case and the special separable case can be connected via Gaussian noise injection. In practice, data augmentation is an effective way to improve robustness and the simplest way is Gaussian noise injection [44]. In this section, we only consider it as an auxiliary tool for theoretical analysis purpose and not for actual robust training. Injecting Gaussian noise amounts to convoluting a Gaussian distribution $N(0, \upsilon^2 \boldsymbol{I}_d)$ to the marginal distribution $P_X$, which enlarges both $\Omega_1$ and $\Omega_2$ to $\mathbb{R}^d$ and a unique decision boundary $\mathcal{D}_\upsilon$ can be induced. Correspondingly, the "noisy" conditional probability, denoted as $\widetilde{\eta}_\upsilon$, is also smoothed to be continuous on $\mathbb{R}^d$. As $\upsilon \to 0$, $\|\widetilde{\eta}_\upsilon - \eta\|_\infty \to 0$ on $\Omega_1$ and $\Omega_2$ and the limiting $\widetilde{\eta}_0$ is a piecewise constant function with discontinuity at the induced decision boundary.

**Lemma 3.5** (Tsybakov's noise condition under Gaussian noises). Let the margin be $2\gamma > 0$, the noise be $N(0, \upsilon^2 \boldsymbol{I}_d)$. Then there exist some constants $T, C > 0$ such that

$$P_X(|2\widetilde{\eta}_\upsilon(X) - 1| < t) \le (C\upsilon^2/\gamma) \exp(-\gamma^2/(2\upsilon^2))t, \forall t \in (0, T).$$

**Theorem 3.6** (Exponential convergence rate). Suppose the classes are separable with margin $2\gamma > 0$. No matter how complicated $\Omega_1 \cup \Omega_2$ are, the excess risk of the over parameterized neural network classifier satisfying Assumptions D.1 and D.4 has the rate $O_\mathbb{P}(e^{-n\gamma/7})$.

Although Theorem 3.6 is similar to that in [45], the techniques are different. Our analysis is directly from the general non-separable case (Theorem 3.1), by the addition of auxiliary noises (Lemma 3.5). The proof of Theorem 3.6 involves taking the auxiliary noise to zero, e.g., $\upsilon = \upsilon_n \asymp 1/\sqrt{n}$. The exponential convergence rate is a direct outcome of Lemma 3.5 and Theorem 3.1. Note that our exponential convergence rate is much faster than existing ones under the similar separable setting [28, 30, 31], which are all polynomial with $n$, e.g., $O_\mathbb{P}(1/\sqrt{n})$.

**Remark 3** (Extension on NTK). Although our analysis only concerns overparameterized one-hidden-layer ReLU neural networks, it can potentially apply to other types of neural networks. Recently, it has been shown that overparameterized multi-layer networks correspond to the Laplace kernel [46, 47]. As long as the trained neural networks can approximate the classifier induced by the NTK, our results can be naturally extended.

**Remark 4.** Theorems 3.4 and 3.6 share the same gist: the overparameterized neural network classifiers can have exponential convergence rate when data are separable with positive margin. The result of Theorem 3.6 is weaker than that of Theorem 3.4, but with milder conditions. Technically, Theorem 3.6 also bridges the non-separable case and separable case through auxiliary noise injection.

### 3.4 Multiclass Classification

In binary classification, the labels are usually encoded as $-1$ and $1$. When there are $K > 2$ classes, the default label coding is one-hot. However, it is empirically observed that this vanilla square loss struggles when the number of classes are large, for which scaling tricks have been proposed [4, 7]. Another popular coding scheme is the simplex coding [11], which takes maximally separated $K$ points on the sphere as label features. When $K = 2$, this reduces to the typical $-1, 1$ coding. Many advantages of the simplex coding have been discussed, including its relationship with cross-entropy loss and supervised contrastive learning [10, 32, 17, 48, 49]. Even though the choice of label coding may have significant impact on optimization, it does not significantly affect our analysis of square loss from a theoretical perspective. In this work, we adopt the simplex coding[3].

Given the label coding, one can easily generalize the theoretical development in Section 3 by employing the following objective function

---

[3]More discussion and empirical comparison about the coding choices can be found in Appendix G.2

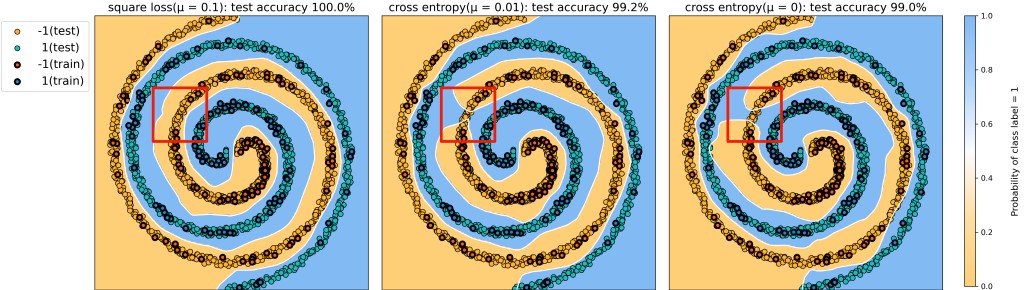

Figure 1: Test misclassification rates and decision boundaries predicted by: SL-ONN + $\ell_2$ (Left); CE-ONN + $\ell_2$ (Center); CE-ONN (Right) in the separable case.

$$\min_{\boldsymbol{W}} \sum_{j=1}^{K} \sum_{i=1}^{n} (f_{j,\boldsymbol{W},\boldsymbol{a}}(\boldsymbol{x}_i) - y_{i,j})^2 + \mu \, \|\boldsymbol{W}\|_2^2,$$

where $f_{\boldsymbol{W},\boldsymbol{a}} : \Omega \mapsto \mathbb{R}^K$, and $\boldsymbol{y}_i = (y_{i,1}, ..., y_{i,K})^\top$ is the label of $i$-th observation.

Next proposition states a relationship between the simplex coding and the conditional probability, which is a generalization of the binary classification case.

**Proposition 3.7** (Conditional probability). Let $f^* : \Omega \to \mathbb{R}^K$ minimize the mean square error $\mathbb{E}_X (f^*(X) - \boldsymbol{v}_y)^2$, where $\boldsymbol{v}_y$ is the simplex coding vector of label $y$. Then

$$\eta_k(\boldsymbol{x}) := \mathbb{P}(y = k|\boldsymbol{x}) = \left((K-1)f^*(\boldsymbol{x})^\top \boldsymbol{v}_k + 1\right)/K. \tag{3.2}$$

Based on the relationship established by Proposition 3.7, one can show that the calibration error can be small in the multi-class classification case. Unlike the softmax function when using cross entropy, the estimated conditional probability using square loss is not guaranteed to be within 0 and 1. This will cause issues for adversarial attacks; see more discussions in Appendix G.2.

## 4 Numerical experiments

Although our theoretical results are for overparameterized neural network in the NTK regime, we expect our conclusions to generalize to practical network architectures. The focus of this section is not on improving the state-of-the-art performance for deep classifiers, but to illustrate the difference between cross-entropy and square loss. We provide experiment results on both synthetic and real data, to support our theoretical findings and illustrate the practical benefits of square loss in training overparameterized DNN classifiers. Compared with cross-entropy, the square loss has comparable generalization performance, but with stronger robustness and smaller calibration error.

### 4.1 Synthetic data

We consider the square loss based, cross-entropy based overparameterized neural networks (ONN) with $\ell_2$ regularization, and the cross-entropy based ONN without $\ell_2$ regularization, denoted as SL-ONN + $\ell_2$, CE-ONN + $\ell_2$, and CE-ONN respectively. For the separable case, we consider two separated classes with spiral curve like supports. Figure 1 shows one instance of the test misclassification rate and decision boundaries attained by three methods, where we can see that SL-ONN + $\ell_2$ has a smaller test misclassification rate and a much smoother decision boundary. In particular, in the red region, where the training data are sparse, SL-ONN + $\ell_2$ fits the correct data distribution best. For the non-separable case, we consider the calibration performance of SL-ONN + $\ell_2$ and CE-ONN + $\ell_2$, where the classifiers are denoted by $\widehat{f}_{l2}$ and $\widehat{f}_{ce}$, respectively. The results presented in Figure G.9 in the Appendix shows that $\widehat{f}_{l2}$ has the smaller mean and standard deviation than $\widehat{f}_{ce}$. More implementation details are in Appendix G.1.

## 4.2 Real data

To make a fair comparison, we adopt popular architectures, ResNet [50] and Wide ResNet [51] and evaluate them on the CIFAR image classification datasets [52], with only the training loss function changed, from cross-entropy (CE) to square loss with simplex coding (SL). Further, we don't employ any large scale hyper-parameter tuning and all the parameters are kept as default except for the learning rate (lr) and batch size (bs), where we are choosing from the better of (lr=0.01, bs=32) and (lr=0.1, bs=128). Each experiment setting is replicated 5 times and we report the average performance followed by its standard deviation in the parenthesis. (lr=0.01, bs=32) works better for the most cases except for square loss trained WRN-16-10 on CIFAR-100. More experiment details and additional results can be found in Appendix G.2.

**Generalization**   In both CIFAR-10 and CIFAR-100, the performance of cross-entropy and square loss with simplex coding are quite comparable, as observed in [4]. Cross-entropy tends to perform slightly better for ResNet, especially on CIFAR-100 with an advantage of less than 1%. There is a more significant gap with Wide ResNet where square loss outperforms cross-entropy by more than 1% on both CIFAR-10 and CIFAR-100. The details can be found in Table 1.

Table 1: Test accuracy on CIFAR datasets. Average accuracy larger than 0 but less than 0.1 is denoted as 0* without standard deviation.

| Dataset | Network | Loss | Clean acc % | PGD-100 ($l_\infty$-strength) | | | AutoAttack ($l_\infty$-strength) | | |
|---------|---------|------|-------------|-------|-------|-------|-------|-------|-------|
| | | | | 2/255 | 4/255 | 8/255 | 2/255 | 4/255 | 8/255 |
| CIFAR-10 | ResNet-18 | CE | **95.15 (0.11)** | 8.81 (1.61) | 0.65 (0.24) | 0 | 2.74 (0.09) | 0 | 0 |
| | | SL | 95.04 (0.07) | **30.53 (0.92)** | **6.64 (0.67)** | **0.86 (0.24)** | **4.10 (0.50)** | **0*** | 0 |
| | WRN-16-10 | CE | 93.94 (0.16) | 1.04 (0.10) | 0 | 0 | 0.33 (0.06) | 0 | 0 |
| | | SL | **95.02 (0.11)** | **37.47 (0.61)** | **23.16 (1.28)** | **7.88 (0.72)** | **5.37 (0.50)** | **0*** | 0 |
| CIFAR-100 | ResNet-50 | CE | **79.82 (0.14)** | 2.31 (0.07) | 0* | 0 | 0.99 (0.10) | 0* | 0 |
| | | SL | 78.91 (0.14) | **13.76 (1.30)** | **4.63 (1.20)** | **1.21 (0.80)** | **3.67 (0.60)** | **0.16 (0.05)** | 0 |
| | WRN-16-10 | CE | 77.89 (0.21) | 0.83 (0.07) | 0* | 0 | 0.42 (0.07) | 0 | 0 |
| | | SL | **79.65 (0.15)** | **6.48 (0.40)** | **0.42 (0.04)** | **0*** | **2.73 (0.20)** | **0*** | 0 |

**Adversarial robustness**   Naturally trained deep classifiers are found to be adversarially vulnerable and adversarial attacks provide a powerful tool to evaluate classification robustness. For our experiment, we consider the black-box Gaussian noise attack, the classic white-box PGD attack [53] and the state-of-the-art AutoAttack [54], with attack strength level 2/255, 4/255, 8/255 in $l_\infty$ norm. AutoAttack contains both white-box and black-box attacks and offers a more comprehensive evaluation of adversarial robustness. The Gaussian noises results are presented in Table G.3 in the Appendix. At different noise levels, square loss consistently outperforms cross-entropy, especially for WRN-16-10, with around 2-4% accuracy improvement. More details can be found in Appendix G.2. The PGD and AutoAttack results are reported in Table 1. Even though classifiers trained with square loss is far away from adversarially robust, it consistently gives significantly higher adversarial accuracy. The same margin can be carried over to standard adversarial training as well. Table 2 lists results from standard PGD adversarial training with CE and SL. By substituting cross-entropy loss to square loss, the robust accuracy increased around 3% while maintaining higher clean accuracy.

One thing to notice is that when constructing white-box attacks, square loss will not work well since it doesn't directly reflect the classification accuracy. More specifically, for a correctly classified image $(\boldsymbol{x}, y)$, maximizing the square loss may result in linear scaling of the classifier $f(\boldsymbol{x})$, which doesn't change the predicted class (see Appendix G.2 for more discussion). To this end, we consider a special attack for classifiers trained by square loss by maximizing the cosine similarity between $f(\boldsymbol{x})$ and $\boldsymbol{v}_y$. We call this angle attack and also utilize it for the PGD adversarial training paired with square loss in Table 2. In our experiments, this special attack rarely outperforms the standard PGD with cross-entropy and the reported PGD accuracy are from the latter settings. This property of square loss may be an advantage in defending adversarial attacks.

**Model calibration**   The predicted class probabilities for square loss can be obtained from Equation 3.2. Expected calibration error (ECE) measures the absolute difference between predicted confidence and actual accuracy. Deep classifiers are usually over-confident [55]. Using ResNet as an example, we report the typical reliability diagram in Figure 2. On CIFAR-10 with ResNet-18,

Table 2: Performance on CIFAR-10 dataset for ResNet-18 under standard PGD adversarial training.

| Loss | Acc (%) | PGD steps | Strength($l_\infty$) | AutoAttack |
|------|---------|-----------|---------------------|------------|
| CE | 86.87 | 3 | 8/255 | 37.08 |
| | 84.50 | 7 | 8/255 | 41.88 |
| SL | **87.31** | 3 | 8/255 | **40.46** |
| | **84.52** | 7 | 8/255 | **44.76** |

the average ECE for cross-entropy is 0.028 (0.002) while that for square loss is 0.0097 (0.001). On CIFAR-100 with ResNet-50, the average ECE for cross-entropy is 0.094 (0.005) while that for square loss is 0.068 (0.005). Square loss results are much more calibrated with significantly smaller ECE.

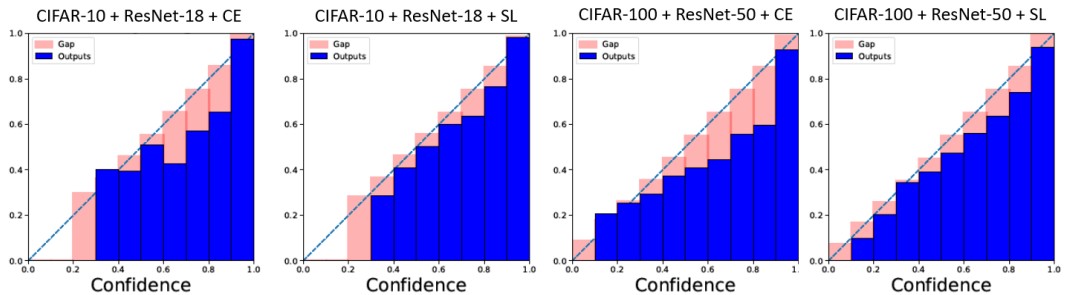

Figure 2: Reliability diagrams of ResNet-18 on CIFAR-10 and ResNet-50 on CIFAR-100. Square loss trained models behave more well-calibrated while cross-entropy trained ones tend to be visibly more over-confident.

## 5   Conclusions

Classification problems are ubiquitous in deep learning. As a fundamental problem, any progress in classification can potentially benefit numerous relevant tasks. Despite its lack of popularity in practice, square loss has many advantages that can be easily overlooked. Through both theoretical analysis and empirical studies, we identify several ideal properties of using square loss in training neural network classifiers, including provable fast convergence rates, strong robustness, and small calibration error. We encourage readers to try square loss in your own application scenarios.

## 6   Acknowledgements

We would like to thank reviewers for their valuable comments on the manuscript. Wenjia Wang and Jun Wang were supported by Foshan HKUST Projects FSUST20-FYTRI03B. We gratefully acknowledge the support of MindSpore for this research.

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
