# Appendix

## A   Gradient Descent and Neural Tangent Kernel

**Gradient Descent**   Since we consider the square loss and $\ell_2$ regularization, the optimization problem in Equation 2.2 becomes

$$\min_{\boldsymbol{W}} \sum_{i=1}^{n} (f_{\boldsymbol{W},\boldsymbol{a}}(\boldsymbol{x}_i) - y_i)^2 + \mu \left\| \boldsymbol{W} \right\|_2^2. \tag{A.1}$$

We consider the GD training of Equation A.1. Let

$$\Phi(\boldsymbol{W}) = \sum_{i=1}^{n} (f_{\boldsymbol{W},\boldsymbol{a}}(\boldsymbol{x}_i) - y_i)^2 + \mu \left\| \boldsymbol{W} \right\|_2^2$$

be the objective function in Equation A.1. The gradient of $\Phi$ with respect to $\boldsymbol{w}_r$ can be written as [38]

$$\frac{\partial \Phi(\boldsymbol{W})}{\partial \boldsymbol{w}_r} = \frac{2}{\sqrt{m}} a_r \sum_{i=1}^{n} (u_i - y_i) \mathbb{I}_{r,i} \boldsymbol{x}_i + 2\mu \boldsymbol{w}_r, \quad r \in [m],$$

where $u_i = f_{\boldsymbol{W},\boldsymbol{a}}(\boldsymbol{x}_i)$ and $\mathbb{I}_{r,i} = \mathbb{I}\{\boldsymbol{w}_r^\top \boldsymbol{x}_i \geq 0\}$. Then the GD update rule is

$$\boldsymbol{w}_r(k+1) = \boldsymbol{w}_r(k) - \zeta \frac{\partial \Phi(\boldsymbol{W})}{\partial \boldsymbol{w}_r} \bigg|_{\boldsymbol{W} = \boldsymbol{W}(k)},$$

where $\boldsymbol{W}(k)$ is the weight matrix at iteration $k$, and $\zeta$ is the learning rate. Define $\mathbb{I}_{r,i}(k) = \mathbb{I}\{\boldsymbol{w}_r(k)^\top \boldsymbol{x}_i \geq 0\}$, $\boldsymbol{Z}(k) \in \mathbb{R}^{md \times n}$ as

$$\boldsymbol{Z}(k) = \frac{1}{\sqrt{m}} \begin{pmatrix} a_1 \mathbb{I}_{1,1}(k)\boldsymbol{x}_1 & \dots & a_1 \mathbb{I}_{1,n}(k)\boldsymbol{x}_n \\ \vdots & \ddots & \vdots \\ a_m \mathbb{I}_{m,1}(k)\boldsymbol{x}_1 & \dots & a_m \mathbb{I}_{m,n}(k)\boldsymbol{x}_n \end{pmatrix},$$

$\boldsymbol{H}(k) = \boldsymbol{Z}(k)^\top \boldsymbol{Z}(k)$, and $\boldsymbol{u}(k) = (\boldsymbol{u}_1(k), ..., \boldsymbol{u}_n(k))^\top$ with $\boldsymbol{u}_i(k) = f_{\boldsymbol{W}(k),\boldsymbol{a}}(\boldsymbol{x}_i)$. Then the GD update rule with respect to $\boldsymbol{W}$ can be written as

$$\mathrm{vec}(\boldsymbol{W}(k+1)) = \mathrm{vec}(\boldsymbol{W}(k)) - 2\zeta \big( \boldsymbol{Z}(k)(\boldsymbol{u}(k) - \boldsymbol{y}) + \mu \mathrm{vec}(\boldsymbol{W}(k)) \big), \tag{A.2}$$

where $\mathrm{vec}(\boldsymbol{W}) = (\boldsymbol{w}_1^\top, \cdots, \boldsymbol{w}_m^\top)^\top \in \mathbb{R}^{md \times 1}$ is the vectorized weight matrix and $\boldsymbol{y} = (y_1, ..., y_n)^\top$.

**Neural Tangent Kernel (NTK)**   It has been shown that the following NTK

$$h(\boldsymbol{s},\boldsymbol{t}) = \mathbb{E}_{\boldsymbol{w} \sim N(0,\boldsymbol{I}_d)} \big( \boldsymbol{s}^\top \boldsymbol{t} \, \mathbb{I}\{\boldsymbol{w}^\top \boldsymbol{s} \geq 0, \boldsymbol{w}^\top \boldsymbol{t} \geq 0\} \big) = \frac{\boldsymbol{s}^\top \boldsymbol{t}(\pi - \arccos(\boldsymbol{s}^\top \boldsymbol{t}))}{2\pi} \tag{A.3}$$

plays an important role in the study of one-hidden-layer ReLU neural networks, where $\boldsymbol{s}, \boldsymbol{t}$ are $d$-dimensional vectors [56, 40]. Since $h$ is positive definite on the unit sphere $\mathbb{S}^{d-1}$ [39], by Mercer's theorem, it possesses a Mercer decomposition as $h(\boldsymbol{s},\boldsymbol{t}) = \sum_{j=0}^{\infty} \lambda_j \varphi_j(\boldsymbol{s})\varphi_j(\boldsymbol{t})$, where $\lambda_1 \geq \lambda_2 \geq ... \geq 0$ are the eigenvalues, and $\{\varphi_j\}_{j=1}^{\infty}$ is an orthonormal basis. The asymptotic behavior of the eigenvalues is described in the following lemma.

**Lemma A.1** (Lemma 3.1 of [40]). Let $\lambda_j$ be the eigenvalues of NTK $h$ defined above. Then we have $\lambda_j \asymp j^{-\frac{d}{d-1}}$.

Let $\mathcal{N}$ denote the reproducing kernel Hilbert space (RKHS) generated by $h$ on $\mathbb{S}^{d-1}$, equipped with norm $\|\cdot\|_{\mathcal{N}}$. As a corollary of Lemma A.1, it can be shown that the ($L_\infty$) entropy number of a unit ball in $\mathcal{N}$, denoted by $\mathcal{N}(1)$, can be controlled. The relationship between the eigenvalues and entropy numbers has been well studied; see [57].

**Lemma A.2.** The entropy number of $\mathcal{N}(1)$, denoted by $H(\mathcal{N}(1), \delta, \|\cdot\|_{L_\infty})$, is bounded by $H(\mathcal{N}(1), \delta, \|\cdot\|_{L_\infty}) \lesssim \delta^{-2(d-1)/d}$.

There are extensive works studying the generalization error bounds under NTK regime. For regression, [45, 40] show the optimal convergence rates when using overparameterized one-hidden-layer neural networks, where the square loss is used. [38] provides generalization error bounds and provable learning scenarios for noiseless data. In the NTK regime, the neural network as a regressor is linked with the nonparametric regression via NTK. There are also other works studying the generalization performance of the neural network as a nonparametric regressor, out of the NTK regime; see [58, 59]. For classification, most of the existing results are established based on the separable data; see [28, 30, 60] and references therein. In particular, [29] consider classification with noisy labels (labels are randomly flipped) and propose to use the square loss with $\ell_2$ regularization. Besides the generalization error bounds, another important research direction is to bridge the gap between NTK and finite-width overparameterized neural networks via GD training; see [56, 38, 61, 40], among others.

## B  Overview of Reproducing Kernel Hilbert Space

We provide here a brief overview of reproducing kernel Hilbert space (RKHS).

**Definition B.1** (Positive Definite Kernel). A function $k : \Omega \times \Omega \mapsto \mathbb{R}$ is said to be a *positive definite kernel*, if $k(\boldsymbol{x}, \widetilde{\boldsymbol{x}}) = k(\widetilde{\boldsymbol{x}}, \boldsymbol{x})$ for all $\boldsymbol{x}, \widetilde{\boldsymbol{x}} \in \Omega$, and

$$\sum_{i=1}^{n} \sum_{j=1}^{n} \beta_i \beta_j k(\boldsymbol{x}_i, \boldsymbol{x}_j) > 0,$$

for all $n \in \mathbb{N}$, $\beta_1, \ldots, \beta_n \in \mathbb{R}$ such that at least one $\beta_j \neq 0$, and $\boldsymbol{x}_1, \ldots, \boldsymbol{x}_n \in \Omega$.

For a positive definite kernel $k$, define a linear space

$$\mathcal{N}_k^0 := \left\{ \sum_{i=1}^{n} \beta_i k(\cdot, \boldsymbol{x}_i) : n \in \mathbb{N}, \ \beta_1, \ldots, \beta_n \in \mathbb{R}, \ \boldsymbol{x}_1, \ldots, \boldsymbol{x}_n \in \Omega \right\},$$

and equip this space with an inner product $\langle \cdot, \cdot \rangle_{\mathcal{N}_k^0}$ by

$$\left\langle \sum_{i=1}^{n} \beta_i k(\cdot, \boldsymbol{x}_i), \sum_{j=1}^{\widetilde{n}} \widetilde{\beta}_j k(\cdot, \widetilde{\boldsymbol{x}}_i) \right\rangle_{\mathcal{N}_k^0} := \sum_{i=1}^{n} \sum_{j=1}^{\widetilde{n}} \beta_i \widetilde{\beta}_j k(\boldsymbol{x}_i, \widetilde{\boldsymbol{x}}_j).$$

The norm of $g \in \mathcal{N}_k^0$ is defined by $\|g\|_{\mathcal{N}_k^0}^2 := \langle g, g \rangle_{\mathcal{N}_k^0}$. Then the RKHS induced by $k$, denoted by $\mathcal{N}_k(\Omega)$, is defined as the closure of $\mathcal{N}_k^0(\Omega)$ with respect to the norm $\|\cdot\|_{\mathcal{N}_k^0(\Omega)}$.

For a subset $\Omega_0 \subset \Omega$, define the *restriction* of $\mathcal{N}_k$ on $\Omega_0$ as

$$\mathcal{N}_k(\Omega_0) := \left\{ g : \Omega_0 \mapsto \mathbb{R} : g = h|_{\Omega_0} \text{ for some } h \in \mathcal{N}_k \right\},$$

where $g = h|_{\Omega_0}$ means $g(\boldsymbol{x}) = h(\boldsymbol{x})$ for all $\boldsymbol{x} \in \Omega_0$. We equip $\mathcal{N}_k(\Omega_0)$ with norm

$$\|g\|_{\mathcal{N}_k(\Omega_0)} := \inf_{\{h \in \mathcal{N}_k : h|_{\Omega_0} = g\}} \|h\|_{\mathcal{N}_k}.$$

Then, $\mathcal{N}_k(\Omega_0)$ is a RKHS with norm $\|\cdot\|_{\mathcal{N}_k(\Omega_0)}$ (see Aronszajn [62, page 351]).

## C  Simplex Coordinates

In simplex label coding, the one-hot labels are replaced by the simplex vertices of a $(K-1)$-simplex. The vertices of a regular $(K-1)$-simplex centered on the origin can be written as:

$$\boldsymbol{v}_0 = \frac{1}{\sqrt{2K}} \cdot (1, \ldots, 1)$$

and for $1 \leq i \leq K-1$,

$$\boldsymbol{v}_i = \frac{1}{\sqrt{2}} \boldsymbol{e}_i - \frac{1}{(K-1)\sqrt{2}} \left( 1 + \frac{1}{\sqrt{K}} \right) \cdot (1, \ldots, 1).$$

The pairwise angle between vertices is $\arccos(-1/(K-1))$ and as $K \to \infty$, the angle converges to $90°$.

The vertices of a $(K-1)$-simplex can be viewed as maximally separated $K$ points on a sphere. In theory, the radius of the sphere doesn't matter but in practice, we recommend scaling it for larger number of classes, e.g., radius = $K$ for $K$-class classification. We find that such scaling empirically outperforms the default radius 1 in our experiments. More details can be found in Appendix G.2.

## D  Assumptions

In this work, we impose the following assumptions. In the rest of the Appendix, we use $\mathrm{poly}(t_1, t_2, \ldots)$ to denote some polynomial function with arguments $t_1, t_2, \ldots$.

**Assumption D.1.** Let $\lambda_{\min}(\boldsymbol{H}^\infty)$ be the minimum eigenvalue of the symmetric matrix $\boldsymbol{H}^\infty$, where $\boldsymbol{H}^\infty = (h(\boldsymbol{x}_i, \boldsymbol{x}_j))_{n \times n}$ ($\boldsymbol{H}^\infty$ is usually called the NTK matrix). Let $\lambda_0$ be the largest number such that with probability at least $1 - \delta_n$, $\lambda_{\min}(\boldsymbol{H}^\infty) \geq \lambda_0$, and $\delta_n \to 0$ as $n$ goes to infinity[4]. For sufficiently large $n$, the regularization parameter $\mu \asymp n^{\frac{d-1}{2d-1}}$, the learning rate $\zeta = o(n^{-\frac{3d-1}{2d-1}})$, the variance of initialization $\xi^2 = O(1)$, the number of nodes in the hidden layer $m \geq \xi^{-2}\mathrm{poly}(n, \lambda_0^{-1})$, and the iteration number $k$ satisfies $\log\left(\mathrm{poly}_1(n, \xi, 1/\lambda_0)\right) \lesssim \zeta\mu k \lesssim \log\left(\mathrm{poly}_2(\xi, 1/n, \sqrt{m})\right)$.

**Assumption D.2.** The conditional probability in the non-separable case satisfies $\eta \in \mathcal{N}$.

**Assumption D.3.** The solution to Equation 2.2 satisfies $\left\|f_{\boldsymbol{W}(k), \boldsymbol{a}}\right\|_{\mathcal{N}} \leq C$, where $C$ is a constant not depending on $n$.

**Remark 5.** Assumption D.3 can be replaced by an alternative assumption, that is, $f_{\boldsymbol{W}(k), \boldsymbol{a}}$ has a bounded Lipschitz constant, and the constant does not depend on $n$.

**Assumption D.4.** The probability density function of the marginal distribution $P_X$, denoted by $p(\boldsymbol{x})$, is continuous on $\Omega$, and there exists a positive constant $c_0$ such that

$$p(\boldsymbol{x}) \leq c_0, \forall \boldsymbol{x} \in \Omega.$$

**Assumption D.5.** The probability density function of the marginal distribution $p(\boldsymbol{x})$ is continuous on $\Omega$, and there exist positive constants $c_1 \leq c_2$ such that

$$c_1 \leq p(\boldsymbol{x}) \leq c_2, \forall \boldsymbol{x} \in \Omega.$$

Assumption D.1 is related to the neural network and GD training, where similar settings have been adopted by [38, 40]. From the results in [38, 40], the width of the neural network depends on the minimum eigenvalue of the NTK matrix $\lambda_{\min}(\boldsymbol{H}^\infty)$, where a smaller $\lambda_{\min}(\boldsymbol{H}^\infty)$ leads to a wider neural network. Therefore, it is desired that $\lambda_0$ is as large as possible. However, the consistency requires that the probability is tending to one; thus, we require $\delta_n \to 0$ as $n \to \infty$. As $n$ becomes larger, with probability tending to one, the distance of the two nearest points in $n$ input points converges to zero, thus making $\boldsymbol{H}^\infty$ close to a degenerate matrix, and the minimum eigenvalue of $\boldsymbol{H}^\infty$ converges to zero. Therefore, inevitably, $\lambda_0 \to 0$ (but $\lambda_{\min}(\boldsymbol{H}^\infty)$ is strictly larger than 0 for all $n$ with probability one). The requirements of the regularization parameter, the learning rate, the variance of initialization, the number of nodes in the hidden layer and the iteration number are all the same as those in [40].

Assumption D.2 imposes conditions on the underlying true conditional probability in the *non-separable* case. This assumption basically requires that the conditional probability is within the function class generated by the GD-trained neural networks we consider (thus can be calibrated). Given that the neural networks are highly flexible, we believe that most of the functions are within the function class generated by the neural networks.

Assumption D.3 is a technical assumption, which requires that the solution to Equation 2.2 is well-behaved, i.e., the solution is within a ball in $\mathcal{N}$ with a certain radius. Roughly speaking, Assumption D.3 requires that the *complexity* of the neural network estimator generated by the GD training is controlled. Since the step size is relatively small and the iteration number is not large (only

---

[4]Potential dependency of $\lambda_0$ on $n$ is suppressed for notational simplicity.

$\log(\text{poly}(n, \xi, , \lambda_0^{-1}))$, we believe it is a mild assumption. It is worth noting that although $f_{\mathbf{W}(k),\mathbf{a}}$ is close to the solution of Equation E.1 under the $L_2$ metric, we cannot confirm whether this closeness still holds under a stronger RKHS metric. Therefore, even if the RKHS norm of the solution of Equation E.1 is bounded, whether the RKHS norm of $f_{\mathbf{W}(k),\mathbf{a}}$ is bounded remains unknown. Thus, we make it as an assumption, and leave it as a future work. Nevertheless, we point out that Theorem 3.1 does not depend on Assumption D.3.

Assumption D.4 only requires the probability to be upper bounded from infinity, while Assumption D.5 requires the probability to be upper bounded from infinity and lower bounded away from zero on the support $\Omega$. They are standard assumptions used in the classical analysis of classification in statistics; see [21, 22] for example. Clearly, uniform distribution satisfies Assumptions D.4 and D.5. In [21], Assumption D.4 is called mild density assumption and Assumption D.5 is called strong density assumption.

# E   Proofs of Main Results

This section includes the proofs of main results in the paper.

## E.1   Proof of Theorem 3.1

We first introduce some lemmas that are used in the proof of Theorem 3.1.

Let $l_1(y_i, f(\boldsymbol{x}_i)) = (1 - y_i f(\boldsymbol{x}_i))^2 = (y_i - f(\boldsymbol{x}_i))^2$ be the square loss on a training point $(\boldsymbol{x}_i, y_i)$, the $l_1$-risk of $f$ be $\mathcal{R}_{l_1}(f) = \mathbb{E}_{X,Y \sim P} l_1(Y, f(X))$, and $\mathcal{R}_{l_1} = \min_{f \in \mathcal{N}} \mathbb{E}_{X,Y \sim P} l_1(Y, f(X))$. Let $L_1(f, \boldsymbol{x}, y) = \mu \|f\|_{\mathcal{N}}^2 + l_1(y, f(\boldsymbol{x}))$ and the $L_1$-risk of $f$ be $\mathcal{R}_{L_1}(f) = \mathbb{E}_{X,Y \sim P} L_1(f, X, Y)$. Let $f_n = \arg\min_{f \in \mathcal{N}} \mathcal{R}_{L_1}(f)$.

Lemma E.1 is (a weaker version of) Theorem 5.6 of [23], which provides a bound on the deviation between the empirical minimizer and true minimizer. Lemma E.2 is used to verify that one of the conditions of Lemma E.1 is fulfilled. Lemma E.3 shows that under certain conditions, the solution to

$$\min_{f \in \mathcal{N}} \frac{1}{n} \sum_{i=1}^{n} (y_i - f(\boldsymbol{x}_i))^2 + \frac{\mu}{n} \|f\|_{\mathcal{N}}^2 \tag{E.1}$$

is closely related to the estimator given by the overparameterized neural networks $f_{\mathbf{W}(k),\boldsymbol{a}}$. Lemma E.3 can be obtained by merely repeating the proof of Theorem 5.2 of [40], since we require that the probability density function $p(\boldsymbol{x})$ of $P_X$ is upper bounded by a positive constant by Assumption D.4. Therefore, the only difference is that we replace $\|\cdot\|_2$ (which corresponds to the uniform distribution) to the $L_2$ norm corresponding to the probability measure $P_X$; thus the proof is omitted. Note also that the second statement of Lemma E.3 corresponds to the noiseless case.

**Lemma E.1.** Let $Z = \Omega \times \{-1, 1\}$. Let $\mathcal{F}$ be a convex set of bounded measurable functions from $Z$ to $\mathbb{R}$ and let $L : \mathcal{F} \times Z \to [0, \infty)$ be a convex and continuous loss function. For a probability measure $P$ on $Z$, define

$$\mathcal{G} := \{L \circ f - L \circ f_{P,\mathcal{F}} : f \in \mathcal{F}\},$$

where $f_{P,\mathcal{F}}$ is a minimizer of $\mathbb{E}_{Z \sim P} L(f, Z)$. Suppose that there are constants $c \geq 0$, $0 < \alpha \leq 1$, $\delta \leq 0$ and $B > 0$ such that $\mathbb{E}_{Z \sim P} g^2 \leq c(\mathbb{E}_{Z \sim P} g)^\alpha + \delta$ and $\|g\|_\infty \leq B$ for all $g \in \mathcal{G}$. Furthermore, assume that $\mathcal{G}$ is separable with respect to $\|\cdot\|_\infty$ and that there are constants $a \geq 1$ and $0 < \alpha < 2$ with

$$\sup_{T \in Z^n} H(B^{-1}\mathcal{G}, \epsilon, \|\cdot\|_{L_2(T)}) \leq a\epsilon^{-\beta} \tag{E.2}$$

for all $\epsilon > 0$, where $H(B^{-1}\mathcal{G}, \epsilon, \|\cdot\|_{L_2(T)})$ is the entropy number of the set $B^{-1}\mathcal{G}$, and $\|f\|_{L_2(T)}^2 = \frac{1}{n} \sum_{i=1}^{n} f(\boldsymbol{x}_i, y_i)^2$ is the empirical norm. Then there exists a constant $c_\beta > 0$ depending only on $\beta$ such that for all $n \geq 1$ and all $t \geq 1$ we have

$$\mathbb{P}(T \in Z^n : \mathcal{R}_{L,P}(f_{T,\mathcal{F}}) > \mathcal{R}_{L,P}(f_{P,\mathcal{F}}) + c_\beta \varepsilon(n, a, B, c, \delta, t)) \leq e^{-t},$$

where

$$\varepsilon(n, a, B, c, \delta, t)$$
$$= B^{2\beta/(4-2\alpha+\alpha p)} c^{(2-\beta)/(4-2\alpha+\alpha\beta)} \left(\frac{a}{n}\right)^{2/(4-2\alpha+\alpha\beta)} + B^{\beta/2} \delta^{(2-\beta)/4} \left(\frac{a}{n}\right)^{1/2}$$
$$+ B \left(\frac{a}{n}\right)^{2/(2+\beta)} + \sqrt{\frac{\delta t}{n}} + \left(\frac{ct}{n}\right)^{1/(2-\alpha)} + \frac{Bt}{n}, \tag{E.3}$$

and $f_{T,\mathcal{F}}$ is the minimizer with respect to the empirical measure.

**Lemma E.2.** Assume the conditions of Theorem 3.1 hold. Define $C := 8||(2\eta - 1)^{-1}||_{\kappa,\infty} + 32$, where $||\cdot||_{\kappa,\infty}$ is the norm of Lorentz space $L_{\kappa,\infty}$ [63]. Let $\mu > 0$ and $0 < \gamma \leq n^{1/2}\mu^{-1/2}$, then for all $f \in \gamma\mathcal{N}(1)$ we have

$$\mathbb{E}_{X,Y\sim P}(L_1 \circ f - L_1 \circ f_n)^2 \leq C(K\gamma + 1)^2(\mathbb{E}_{X,Y\sim P}(L_1 \circ f - L_1 \circ f_n)) + 2C(K\gamma + 1)^2 a(\mu),$$

where $a(\mu)$ is the approximation error function given by

$$a(\mu) = \inf_{f \in \mathcal{N}}(n^{-1}\mu||f||_{\mathcal{N}}^2 + \mathcal{R}_{l_1}(f) - \mathcal{R}_{l_1}).$$

**Lemma E.3.** Suppose Assumptions D.1 and D.4 hold. Then we have

$$\mathbb{E}_{X\sim P_X}(f_{\boldsymbol{W}(k),\boldsymbol{a}}(X) - \widehat{f}(X))^2 = O_{\mathbb{P}}(n^{-\frac{d}{2d-1}}),$$

where $\widehat{f}$ is the solution to Equation E.1. Furthermore, if there exists a function $f \in \mathcal{N}$ that does not depend on $n$ and $f(\boldsymbol{x}_i) = y_i$ for all $i = 1, ..., n$, then we can set $\mu = o(1)$ and obtain

$$\mathbb{E}_{X\sim P_X}(f_{\boldsymbol{W}(k),\boldsymbol{a}}(X) - \widehat{f}(X))^2 = o_{\mathbb{P}}(1).$$

**Remark 6.** According to the proof in [40], the probability in $o_{\mathbb{P}}(1)$ of Lemma E.3 only relates to the width of the one-hidden-layer neural network, which can be arbitrarily small by enlarging the neural network's width.

Now we are ready to prove Theorem 3.1. Let $L = L_1$ in Lemma E.1, which is clearly continuous. Let $\widehat{f}$ be the solution to Equation E.1. The key idea in this proof is using $\widehat{f}$ to bridge two functions $2\eta - 1$ and $f_{\boldsymbol{W}(k),\boldsymbol{a}}$.

Since $\widehat{f}$ is the solution to Equation E.1, it can be seen that

$$\frac{1}{n}\sum_{i=1}^{n}(y_i - \widehat{f}(\boldsymbol{x}_i))^2 + \frac{\mu}{n}\left\|\widehat{f}\right\|_{\mathcal{N}}^2 \leq \frac{1}{n}\sum_{i=1}^{n}(y_i - (2\eta(\boldsymbol{x}_i) - 1))^2 + \frac{\mu}{n}\|2\eta(\boldsymbol{x}_i) - 1\|_{\mathcal{N}}^2$$

$$\leq \frac{2}{n}\sum_{i=1}^{n}(y_i^2 + (2\eta(\boldsymbol{x}_i) - 1)^2) + \frac{\mu}{n}\|2\eta(\boldsymbol{x}_i) - 1\|_{\mathcal{N}}^2$$

$$\leq C_1, \tag{E.4}$$

where the second inequality is by the Cauchy-Schwarz inequality, and the third inequality is because $y_i^2 = 1$ and $\eta(\boldsymbol{x})$ is bounded.

The reproducing property implies that

$$\widehat{f}(\boldsymbol{x}) = \langle \widehat{f}, h(\boldsymbol{x}, \cdot)\rangle_{\mathcal{N}} \leq \left\|\widehat{f}\right\|_{\mathcal{N}}\|h(\boldsymbol{x}, \cdot)\|_{\mathcal{N}} = \left\|\widehat{f}\right\|_{\mathcal{N}}\sqrt{h(\boldsymbol{x}, \boldsymbol{x})}, \forall \boldsymbol{x} \in \Omega,$$

which yields

$$\left\|\widehat{f}\right\|_{L_\infty} \leq C_2\left\|\widehat{f}\right\|_{\mathcal{N}}.$$

Together with Equation E.4, we obtain

$$\left\|\widehat{f}\right\|_{L_\infty} \leq C_3\left\|\widehat{f}\right\|_{\mathcal{N}} \leq C_4(\mu/n)^{-1/2}. \tag{E.5}$$

Thus, we can take $B = C_4(\mu/n)^{-1/2}$ in Lemma E.1. The entropy condition can be verified via Lemma A.2, which allows us to take $\beta = 2(d-1)/d$. Equation E.5, together with Lemma E.2, also suggests that we can take $c = C(KB+1)^2$, $\alpha = 1$, and $\delta = 2C(KB+1)^2 a(\mu)$.

Next, we provide an upper bound on $a(\mu)$. The definition of $a(\mu)$ implies

$$
\begin{aligned}
a(\mu) =& n^{-1}\mu \|f_n\|_{\mathcal{N}}^2 + R_{l_1}(f_n) - R_{l_1} \\
=& n^{-1}\mu \|f_n\|_{\mathcal{N}}^2 + \mathbb{E}_{X \sim P_X}(2\eta(X) - 1 - f_n(X))^2 \\
\leq& n^{-1}\mu \|2\eta - 1\|_{\mathcal{N}}^2,
\end{aligned}
\tag{E.6}
$$

where we use the relationship $\mathcal{R}_{l_1,P}(f) - \mathcal{R}_{l_1,P} = \mathbb{E}_{X \sim P_X}(2\eta(X) - 1 - f(X))^2$.

Plugging all the terms into Equation E.3, together with Lemma E.1, yields that

$$
\mathcal{R}_{L_1,P}(\widehat{f}) = \mathcal{R}_{L_1,P}(f_n) + O_{\mathbb{P}}(\varepsilon(n,a,B,c,\delta)),
\tag{E.7}
$$

where

$$
\begin{aligned}
\varepsilon(n,a,B,c,\delta) =& B^{\frac{4}{2+\beta}} n^{-\frac{2}{2+\beta}} + B(\mu/n)^{\frac{2-\beta}{4}} n^{-\frac{1}{2}} \|2\eta - 1\|_{\mathcal{N}}^{\frac{2-\beta}{2}} + B^2 n^{-1} \\
=& B^{\frac{4d}{4d-2}} n^{-\frac{2d}{4d-2}} + B\mu^{\frac{1}{2d}} n^{-\frac{1}{2}-\frac{1}{2d}} \|2\eta - 1\|_{\mathcal{N}}^{\frac{1}{d}} + B^2 n^{-1}.
\end{aligned}
\tag{E.8}
$$

Since $\mathcal{R}_{l_1,P}(f) - \mathcal{R}_{l_1,P} = \mathbb{E}_{X \sim P_X}(2\eta(X) - 1 - f(X))^2$, we subtract $\mathcal{R}_{l_1,P}$ on both sides of Equation E.7 and get

$$
\begin{aligned}
& \mathbb{E}_{X \sim P_X}(2\eta(X) - 1 - \widehat{f}(X))^2 + n^{-1}\mu \left\|\widehat{f}\right\|_{\mathcal{N}}^2 \\
=& \mathbb{E}_{X \sim P_X}(2\eta(X) - 1 - f_n(X))^2 + n^{-1}\mu \|f_n\|_{\mathcal{N}}^2 + O_{\mathbb{P}}(\varepsilon(n,a,B,c,\delta)) \\
=& O_{\mathbb{P}}(n^{-1}\mu \|2\eta - 1\|_{\mathcal{N}}^2 + \varepsilon(n,a,B,c,\delta)),
\end{aligned}
\tag{E.9}
$$

where the last equality (with big $O$ notation) is by Equation E.6. Combining Equation E.9 and Equation E.5 implies

$$
\left\|\widehat{f}\right\|_{L_\infty}^2 \leq C_3^2 \left\|\widehat{f}\right\|_{\mathcal{N}}^2 = O_{\mathbb{P}}(1 + n\mu^{-1}\varepsilon(n,a,B,c,\delta)).
$$

In the following, we will show that by taking $\mu \asymp n^{\frac{d-1}{2d-1}}$,

$$
\mathbb{E}_{X \sim P_X}(2\eta(X) - 1 - \widehat{f}(X))^2 + n^{-1}\mu \left\|\widehat{f}\right\|_{\mathcal{N}}^2 = O_{\mathbb{P}}(n^{-\frac{d}{2d-1}} \max(1, \|2\eta - 1\|_{\mathcal{N}}^{\frac{2}{d}})) = O_{\mathbb{P}}(n^{-\frac{d}{2d-1}}).
\tag{E.10}
$$

If $n\mu^{-1}\varepsilon(n,a,B,c,\delta) \lesssim 1$, then $\varepsilon(n,a,B,c,\delta) \lesssim \mu/n$, and Equation E.10 holds. Otherwise, we can replace $B^2$ by its upper bound $O_{\mathbb{P}}(n\mu^{-1}\varepsilon(n,a,B,c,\delta))$ in Equation E.8 and obtain that

$$
\varepsilon = O_{\mathbb{P}}(\varepsilon^{\frac{d}{2d-1}}\mu^{-\frac{d}{2d-1}} + \varepsilon^{\frac{1}{2}}\mu^{-\frac{d-1}{2d}} n^{-\frac{1}{2d}} \|2\eta - 1\|_{\mathcal{N}}^{\frac{1}{d}}),
$$

where we set $\varepsilon = \varepsilon(n,a,B,c,\delta)$ for notational simplicity. Let us hereby denote $I_1 = \varepsilon^{\frac{d}{2d-1}}\mu^{-\frac{d}{2d-1}}$ and $I_2 = \varepsilon^{\frac{1}{2}}\mu^{-\frac{d-1}{2d}} n^{-\frac{1}{2d}} \|2\eta - 1\|_{\mathcal{N}}^{\frac{1}{d}}$, and consider the following two cases.

**Case 1:** $I_1 \geq I_2$, then we have

$$
\varepsilon = O_{\mathbb{P}}(\varepsilon^{\frac{d}{2d-1}}\mu^{-\frac{d}{2d-1}}).
$$

Solving this equality leads to

$$
\varepsilon = O_{\mathbb{P}}(\mu^{-\frac{d}{d-1}}).
\tag{E.11}
$$

Plugging Equation E.11 into Equation E.9 and minimize the right-hand side of Equation E.9 with respect to $\mu$ gives us $\mu \asymp n^{\frac{d-1}{2d-1}}$; thus Equation E.10 holds.

**Case 2:** $I_1 < I_2$, then we have

$$
\varepsilon = O_{\mathbb{P}}(\varepsilon^{\frac{1}{2}}\mu^{-\frac{d-1}{2d}} n^{-\frac{1}{2d}} \|2\eta - 1\|_{\mathcal{N}}^{\frac{1}{d}}),
$$

which leads to

$$\varepsilon = O_{\mathbb{P}}(\mu^{-\frac{d-1}{d}} n^{-\frac{1}{d}} \|2\eta - 1\|_{\mathcal{N}}^{\frac{2}{d}}). \tag{E.12}$$

Similarly, we plug Equation E.12 into Equation E.9 and minimize the right-hand side of Equation E.9 with respect to $\mu$ and obtain $\mu \asymp n^{\frac{d-1}{2d-1}}$, which also leads to Equation E.10.

Now we can obtain an upper bound on the excess risk. For the notation simplicity, let $f = f_{\boldsymbol{W}(k),\boldsymbol{a}}$. The excess risk can be bounded by

$$\begin{aligned} L(f) - L^* \leq & \mathbb{E}_{X \sim P_X} \mathbb{I}\{(2\eta(X) - 1)f(X) \leq 0, |\eta(X) - 0.5| < \delta\} |2\eta(X) - 1| \\ & + \mathbb{E}_{X \sim P_X} \mathbb{I}\{(2\eta(X) - 1)f(X) \leq 0, |\eta(X) - 0.5| \geq \delta\} |2\eta(X) - 1|. \end{aligned} \tag{E.13}$$

The first term can be bounded via Tsybakov's noise condition as

$$\mathbb{E}_{X \sim P_X} \mathbb{I}\{(2\eta(X) - 1)f(X) \leq 0, |\eta(X) - 0.5| < \delta\} |2\eta(X) - 1| \leq 2\delta \mathbb{E}[\mathbb{I}\{|\eta(X) - 0.5| < \delta\}]$$
$$= 2\delta \mathbb{P}(|\eta(X) - 0.5| < \delta) \leq 2C\delta^{\kappa+1}. \tag{E.14}$$

It remains to bound the second term in Equation E.13. If $p(\boldsymbol{x})$ is continuous, then by the fact that $|2\eta(X) - 1| \leq |2\eta(X) - 1 - f(X)|$ if $(2\eta(X) - 1)f(X) \leq 0$, we have

$$\begin{aligned} & \mathbb{E}_{X \sim P_X} \mathbb{I}\{(2\eta(X) - 1)f(X) \leq 0, |\eta(X) - 0.5| \geq \delta\} |2\eta(X) - 1| \\ \leq & 2\delta^{-1} \mathbb{E}_{X \sim P_X} \mathbb{I}\{(2\eta(X) - 1)f(X) \leq 0, |\eta(X) - 0.5| \geq \delta\} |2\eta(X) - 1|^2 \\ \leq & 2\delta^{-1} \mathbb{E}_{X \sim P_X} \mathbb{I}\{|\eta(X) - 0.5| \geq \delta\} (2\eta(X) - 1 - f(X))^2 \\ \leq & 2\delta^{-1} \mathbb{E}_{X \sim P_X} (2\eta(X) - 1 - f(X))^2 \\ \leq & 4\delta^{-1} \mathbb{E}_{X \sim P_X} (2\eta(X) - 1 - \widehat{f}(X))^2 + 4\delta^{-1} \mathbb{E}_{X \sim P_X} (f(X) - \widehat{f}(X))^2 \\ = & O_{\mathbb{P}}(\delta^{-1} n^{-\frac{d}{2d-1}}), \end{aligned} \tag{E.15}$$

where the fourth inequality is by the Cauchy-Schwarz inequality, and the last equality (with big $O$ notation) is by Equation E.10 and Lemma E.3. Taking $\delta = n^{-\frac{d}{(2d-1)(\kappa+2)}}$, and plugging Equation E.14 and Equation E.15 into Equation E.13 leads to

$$L(f) = L^{\star} + O_{\mathbb{P}}(n^{-\frac{d(\kappa+1)}{(2d-1)(\kappa+2)}}). $$

This finishes the proof.

## E.2   Proof of Theorem 3.2

We first present a lemma.

**Lemma E.4.** Suppose two sets are separable with a positive margin $\gamma > 0$. Then there exists a function $f_T$ satisfying

$$f_T(\boldsymbol{x}) = 1, \forall \boldsymbol{x} \in \Omega_1, \quad f_T(\boldsymbol{x}) = -1, \forall \boldsymbol{x} \in \Omega_2.$$

*Proof of Theorem 3.2.* By the equivalence of the RKHS generated by the Laplace kernel and $\mathcal{N}$ [46, 47], it can be shown that $\mathcal{N}$ can be embedded into the Sobolev space $W_2^\nu$ for some $\nu > d/2$. Consider the Hölder space $C_b^{0,\alpha}$ for $0 < \alpha \leq 1$ equipped with the norm

$$\|f\|_{C_b^{0,\alpha}} := \sup_{\boldsymbol{x},\boldsymbol{x}' \in \Omega, \boldsymbol{x} \neq \boldsymbol{x}'} \frac{|f(\boldsymbol{x}) - f(\boldsymbol{x}')|}{\|\boldsymbol{x} - \boldsymbol{x}'\|_2^\alpha}. \tag{E.16}$$

By the Sobolev embedding theorem, we have the embedding relationship

$$\|f\|_{C_b^{0,\tau}} \leq C_1 \|f\|_{W_2^\nu} \leq C_2 \|f\|_{\mathcal{N}} \tag{E.17}$$

for all $f \in \mathcal{N}$, where $\tau = \min(\nu - d/2, 1)$.

Without loss of generality, let us consider $\boldsymbol{x} \in \Omega_1$. The case of $\boldsymbol{x} \in \Omega_2$ can be proved similarly. For any $\boldsymbol{x} \in \Omega_1$, take $\boldsymbol{x}' = \arg\min_{\boldsymbol{x}_i} \|\boldsymbol{x}_i - \boldsymbol{x}\|_2$. Thus, the definition of the Hölder space and Equation E.17 imply

$$|f_{\boldsymbol{W}(k),\boldsymbol{a}}(\boldsymbol{x}) - f_{\boldsymbol{W}(k),\boldsymbol{a}}(\boldsymbol{x}')| \leq C_2 \|f_{\boldsymbol{W}(k),\boldsymbol{a}}\|_{\mathcal{N}} \|\boldsymbol{x}' - \boldsymbol{x}\|_2^\tau \leq C_3 \|\boldsymbol{x}' - \boldsymbol{x}\|_2^\tau, \tag{E.18}$$

where the last inequality is by Assumption D.3.

Let $\widehat{f}$ be the solution to Equation E.1, and let $\widehat{f}_1$ be the solution to Equation E.1 with $\mu = 0$. Let $f_T$ be as in Lemma E.4. Note that $\widehat{f}_1$ satisfies $\widehat{f}_1(\boldsymbol{x}_i) = f_T(\boldsymbol{x}_i)$. Thus, by the identity $\left\|\widehat{f}_1\right\|_{\mathcal{N}}^2 + \left\|\widehat{f}_1 - f_T\right\|_{\mathcal{N}}^2 = \left\|\widehat{f}_T\right\|_{\mathcal{N}}^2$ [64], we have $\left\|\widehat{f}_1\right\|_{\mathcal{N}} \leq \|f_T\|_{\mathcal{N}}$. Since $\widehat{f}$ is the solution to Equation E.1, we have

$$\frac{1}{n}\sum_{i=1}^n (y_i - \widehat{f}(\boldsymbol{x}_i))^2 + \frac{\mu}{n}\left\|\widehat{f}\right\|_{\mathcal{N}}^2 \leq \frac{1}{n}\sum_{i=1}^n (y_i - f_T(\boldsymbol{x}_i))^2 + \frac{\mu}{n}\|f_T\|_{\mathcal{N}}^2 = \frac{\mu}{n}\|f_T\|_{\mathcal{N}}^2, \quad \text{(E.19)}$$

which implies $\left\|\widehat{f}\right\|_{\mathcal{N}} \leq \|f_T\|_{\mathcal{N}}$, where we utilize $y_i = f_T(\boldsymbol{x}_i)$ in the separable case.

Direct computation shows

$$\begin{aligned} f_{\boldsymbol{W}(k),\boldsymbol{a}}(\boldsymbol{x}') &= \widehat{f}_1(\boldsymbol{x}') - (\widehat{f}_1(\boldsymbol{x}') - \widehat{f}(\boldsymbol{x}')) - (\widehat{f}(\boldsymbol{x}') - f_{\boldsymbol{W}(k),\boldsymbol{a}}(\boldsymbol{x}')) \\ &= 1 - I_1 - I_2, \end{aligned} \quad \text{(E.20)}$$

where we use $\widehat{f}_1(\boldsymbol{x}_i) = 1$ for any $\boldsymbol{x}_i \in \Omega$; thus $\widehat{f}_1(\boldsymbol{x}') = 1$.

By the representer theorem, $\widehat{f}$ and $\widehat{f}_1$ can be expressed as

$$\widehat{f}_1(\boldsymbol{x}) = h(\boldsymbol{x}, \boldsymbol{X})(\boldsymbol{H}^\infty + \mu \boldsymbol{I}_n)^{-1}\boldsymbol{y}, \widehat{f}(\boldsymbol{x}) = h(\boldsymbol{x}, \boldsymbol{X})(\boldsymbol{H}^\infty)^{-1}\boldsymbol{y},$$

where $h(\boldsymbol{x}, \boldsymbol{X}) = (h(\boldsymbol{x}, \boldsymbol{x}_1), ..., h(\boldsymbol{x}, \boldsymbol{x}_n)) \in \mathbb{R}^{1 \times n}$, $\boldsymbol{H}^\infty = (h(\boldsymbol{x}_i, \boldsymbol{x}_j))_{n \times n}$, and $\boldsymbol{y} = (y_1, ..., y_n)^\top = (f_T(\boldsymbol{x}_1), ..., f_T(\boldsymbol{x}_n))^\top$. Thus, the first term $I_1$ in Equation E.20 can be bounded by

$$\begin{aligned} |I_1| &= |\widehat{f}_1(\boldsymbol{x}') - \widehat{f}(\boldsymbol{x}')| = |h(\boldsymbol{x}', \boldsymbol{X})(\boldsymbol{H}^\infty)^{-1}\boldsymbol{y} - h(\boldsymbol{x}', \boldsymbol{X})(\boldsymbol{H}^\infty + \mu \boldsymbol{I}_n)^{-1}\boldsymbol{y}| \\ &= |\mu h(\boldsymbol{x}', \boldsymbol{X})(\boldsymbol{H}^\infty)^{-1}(\boldsymbol{H}^\infty + \mu \boldsymbol{I}_n)^{-1}\boldsymbol{y}| \\ &\leq \mu \sqrt{h(\boldsymbol{x}', \boldsymbol{X})(\boldsymbol{H}^\infty)^{-1}(\boldsymbol{H}^\infty + \mu \boldsymbol{I}_n)^{-1}(\boldsymbol{H}^\infty)^{-1}h(\boldsymbol{x}', \boldsymbol{X})^\top \boldsymbol{y}^\top (\boldsymbol{H}^\infty + \mu \boldsymbol{I}_n)^{-1}\boldsymbol{y}} \\ &= \mu \sqrt{h(\boldsymbol{x}', \boldsymbol{X})(\boldsymbol{H}^\infty)^{-1}(\boldsymbol{H}^\infty + \mu \boldsymbol{I}_n)^{-1}(\boldsymbol{H}^\infty)^{-1}h(\boldsymbol{x}', \boldsymbol{X})^\top} \left\|\widehat{f}_1\right\|_{\mathcal{N}} \\ &\leq \sqrt{\mu}\sqrt{h(\boldsymbol{x}', \boldsymbol{X})(\boldsymbol{H}^\infty)^{-2}h(\boldsymbol{x}', \boldsymbol{X})^\top}\|f_T\|_{\mathcal{N}} = \sqrt{\mu}\|f_T\|_{\mathcal{N}}, \end{aligned} \quad \text{(E.21)}$$

where the first inequality is by the Cauchy-Schwarz inequality, the second inequality is because $(\boldsymbol{H}^\infty + \mu \boldsymbol{I}_n)^{-1} \preceq \mu^{-1}\boldsymbol{I}_n$, and the last equality is because for any $\boldsymbol{x}_i$, $(\boldsymbol{H}^\infty)^{-1}h(\boldsymbol{x}_i, \boldsymbol{X})^\top = \boldsymbol{e}_i$. Therefore, $I_1$ converges to zero as $n \to \infty$ since $\mu = o(1)$. Specifically, there exists an $n_1$ such that when $n \geq n_1$, $|I_1| \leq 1/4$.

The second term $I_2$ in Equation E.20 can be bounded by

$$\begin{aligned} |I_2| &\leq \left\|\widehat{f} - f_{\boldsymbol{W}(k),\boldsymbol{a}}\right\|_\infty \leq C_4 \left\|\widehat{f} - f_{\boldsymbol{W}(k),\boldsymbol{a}}\right\|_{\mathcal{N}}^{\frac{d-1}{d}} \left\|\widehat{f} - f_{\boldsymbol{W}(k),\boldsymbol{a}}\right\|_2^{\frac{1}{d}} \\ &\leq C_4 (\left\|\widehat{f}\right\|_{\mathcal{N}} + \|f_{\boldsymbol{W}(k),\boldsymbol{a}}\|_{\mathcal{N}})^{\frac{d-1}{d}} \left\|\widehat{f} - f_{\boldsymbol{W}(k),\boldsymbol{a}}\right\|_2^{\frac{1}{d}} \\ &\leq C_5 \left(\mathbb{E}_{X \sim P_X}(\widehat{f}(X) - f_{\boldsymbol{W}(k),\boldsymbol{a}}(X))^2\right)^{\frac{1}{2d}}, \end{aligned} \quad \text{(E.22)}$$

which converges to zero by Lemma E.3. In Equation E.22, the second inequality is by the interpolation inequality, the third inequality is by the triangle inequality, and the last inequality is because of Assumption D.5. Therefore, there exists an $n_2$ such that when $n \geq n_2$, with probability at least $1 - \delta$, $|I_2| \leq 1/4$.

Take $n_0 = \max(n_1, n_2)$. For $n \geq n_0$, Equation E.20 gives us $f_{\boldsymbol{W}(k),\boldsymbol{a}}(\boldsymbol{x}') \geq 1/2$ with probability at least $1 - \delta$. Therefore, by Equation E.18, as long as

$$\|\boldsymbol{x}' - \boldsymbol{x}\|_2 = \min_{\boldsymbol{x}_i}\|\boldsymbol{x}_i - \boldsymbol{x}\|_2 \leq (4C_3)^{-1/\tau} := C_6, \forall \boldsymbol{x} \in \Omega_1, \quad \text{(E.23)}$$

we have $f_{\boldsymbol{W}(k),\boldsymbol{a}}(\boldsymbol{x}) \geq 1/4$ for all $\boldsymbol{x} \in \Omega_1$, which implies that the missclassification rate is zero.

Let $N(\delta, \Omega_1, \|\cdot\|_2)$ be the covering number of $\Omega_1$ and $N_0 = N(C_6/2, \Omega_1, \|\cdot\|_2)$. Since $\Omega_1$ is compact and $C_6 > 0$, $N_0$ is finite (and is a constant). Therefore, $\Omega_1$ can be covered by $N_0$ balls with radius $C_6/2$ (denoted by $\mathbf{B}$), and as long as for each ball, there exists one point $\boldsymbol{x}_j$ in this ball, Equation E.23 is satisfied. Since $\boldsymbol{x}_k$ has a probability density function with lower bound $c_1$, it remains to upper bound the probability that there exists one ball such that there is no point in it. Define this event as $\mathcal{A}$. The union bound of probability implies that for $n > n_0$,

$$\mathbb{P}(\mathcal{A}) \leq N_0 \left(1 - \frac{c_1 Vol(\mathbf{B})}{Vol(\Omega_1)}\right)^n \leq N_0 \exp(-C_7 n),$$

where $C_7 = -\log\left(1 - \frac{c_1 Vol(\mathbf{B})}{Vol(\Omega_1)}\right)$ is a positive constant. Clearly, we can adjust the constants such that the results in Theorem 3.2 holds for all $n$. This finishes the proof.

### E.3 Proof of Theorem 3.3

Note that $f_{\boldsymbol{W}(k),\boldsymbol{a}}$ is a classifier, and the decision boundary is defined by $\mathcal{D}_T := \{\boldsymbol{x} | f_{\boldsymbol{W}(k),\boldsymbol{a}}(\boldsymbol{x}) = 0\}$. Take any point $\boldsymbol{x}'$ in $\mathcal{D}_T$. The definition of the Hölder space and Equation E.17 imply

$$\frac{|f_{\boldsymbol{W}(k),\boldsymbol{a}}(\boldsymbol{x}) - f_{\boldsymbol{W}(k),\boldsymbol{a}}(\boldsymbol{x}')|}{\|\boldsymbol{x} - \boldsymbol{x}'\|_2^\tau} \leq C_2 \|f_{\boldsymbol{W}(k),\boldsymbol{a}}\|_{\mathcal{N}} \leq C_3, \forall \boldsymbol{x} \in \Omega, \tag{E.24}$$

which is the same as

$$\|\boldsymbol{x} - \boldsymbol{x}'\|_2^\tau \geq |f_{\boldsymbol{W}(k),\boldsymbol{a}}(\boldsymbol{x})|/C_3, \forall \boldsymbol{x} \in \Omega, \tag{E.25}$$

where the last inequality in Equation E.24 is because of Assumption D.3. Therefore, it suffices to provide a lower bound of $|f_{\boldsymbol{W}(k),\boldsymbol{a}}(\boldsymbol{x})|$. Without loss of generality, let $\boldsymbol{x} \in \Omega_1$, because the case $\boldsymbol{x} \in \Omega_2$ can be proved similarly. However, this has already been proved in the proof of Theorem 3.2, where we showed that with probability at least $1 - \delta - C_4 \exp(-C_5 n)$, $f_{\boldsymbol{W}(k),\boldsymbol{a}}(\boldsymbol{x}) \geq 1/4$ for all $\boldsymbol{x} \in \Omega_1$.

### E.4 Proof of Theorem 3.4

By applying the interpolation inequality, the $L_\infty$ norm of $2\eta - 1 - f_{\boldsymbol{W}(k),\boldsymbol{a}}$ can be bounded by

$$\begin{aligned}
\left\|2\eta - 1 - f_{\boldsymbol{W}(k),\boldsymbol{a}}\right\|_\infty &\leq C_0 \left\|2\eta - 1 - f_{\boldsymbol{W}(k),\boldsymbol{a}}\right\|_2^{\frac{1}{d}} \left\|2\eta - 1 - f_{\boldsymbol{W}(k),\boldsymbol{a}}\right\|_{W^\nu}^{\frac{d-1}{d}} \\
&\leq C_1 \left\|2\eta - 1 - f_{\boldsymbol{W}(k),\boldsymbol{a}}\right\|_2^{\frac{1}{d}} \left\|2\eta - 1 - f_{\boldsymbol{W}(k),\boldsymbol{a}}\right\|_{\mathcal{N}}^{\frac{d-1}{d}} \\
&\leq C_2 \left\|2\eta - 1 - f_{\boldsymbol{W}(k),\boldsymbol{a}}\right\|_2^{\frac{1}{d}} \left(\|2\eta - 1\|_{\mathcal{N}} + \|f_{\boldsymbol{W}(k),\boldsymbol{a}}\|_{\mathcal{N}}\right)^{\frac{d-1}{d}} \\
&\leq C_3 \left\|2\eta - 1 - f_{\boldsymbol{W}(k),\boldsymbol{a}}\right\|_2^{\frac{1}{d}} \\
&\leq C_4 \left(\mathbb{E}_{X \sim P_X}(2\eta(X) - 1 - f_{\boldsymbol{W}(k),\boldsymbol{a}}(X))^2\right)^{\frac{1}{2d}} = O_{\mathbb{P}}(n^{-\frac{1}{4d-2}}) \quad \text{(E.26)}
\end{aligned}$$

where the second equality is by the equvilance of the Sobolev space $W^\nu$ and the RKHS $\mathcal{N}$; the third inequality is by the triangle inequality; the fourth inequality is by Assumptions D.2 and D.3; the fifth inequality is because of Assumption D.5; and the last equality (with big $O$ notation) is because of Equation E.15. This finishes the proof.

### E.5 Proof of Lemma 3.5

Let's first consider the simplest $d = 1$ case where $\Omega_1 = \{\gamma\}$ and $\Omega_2 = \{-\gamma\}$. Let $\phi$ denote the standard normal $N(0, 1)$ density. By injecting Gaussian noises $N(0, \upsilon^2)$, the induced conditional probability can be written as

$$\widetilde{\eta}_\upsilon(x) = \frac{\phi(\frac{x-\gamma}{\upsilon})}{\phi(\frac{x-\gamma}{\upsilon}) + \phi(\frac{x+\gamma}{\upsilon})} = \frac{1}{1 + \exp(-\frac{2\gamma x}{\upsilon^2})}.$$

For small enough $1/2 > t > 0$, direct calculation yields $\{x \in \mathbb{R} : |2\widetilde{\eta}_\upsilon(x) - 1| < t\} = (-x_t, x_t)$ where

$$x_t = \frac{\upsilon^2}{2\gamma} \log\left(\frac{1+t}{1-t}\right) \leq \frac{2\upsilon^2}{\gamma} t.$$

Hence,

$$P_X(|2\widetilde{\eta}_v(x) - 1| < t) = P_X(-x_t < x < x_t) \le 2x_t \phi((\gamma + x_t)/v)$$
$$\le \frac{Cv^2}{\gamma} \exp\left(-\frac{\gamma^2}{2v^2}\right) \cdot t.$$

In general cases, notice that Tsybakov's noise condition measures the separation between classes. Therefore, the bottleneck for the inequality is where $\Omega_1$ and $\Omega_2$ are the closest, i.e., where margin $2\gamma$ is attained. Let $\boldsymbol{x}_+ \in \Omega_1$ and $\boldsymbol{x}_- \in \Omega_2$ satisfy $\|\boldsymbol{x}_+ - \boldsymbol{x}_-\|_2 = 2\gamma$ (which can be attained since $\Omega$ is compact). Consider the delta distribution at $\boldsymbol{x}_+$ and $\boldsymbol{x}_-$, which is less separated than the original distribution. Then, it reduces to the simplest case.

### E.6 Proof of Theorem 3.6

A closer look at the proof of Theorem 3.1 reveals that the convergence rate depends polynomially on the constant in Tsybakov's noise condition. Specifically, it can be checked that $\|\widetilde{\eta}_v\|_{\mathcal{N}} = O(\text{poly}(1/v))$. Let $D^\alpha f := \frac{\partial^{|\alpha|}}{\partial x_1^{\alpha_1} \cdots x_d^{\alpha_d}} f$ denote the $\alpha$-th (weak) derivative of a function $f$ with $|\alpha| = \alpha_1 + \ldots + \alpha_d$ for a multi-index $\alpha = (\alpha_1, \cdots, \alpha_d) \in \mathbb{N}_0^d$. The Sobolev embedding theorem implies that $\|\widetilde{\eta}_v\|_{\mathcal{N}}$ is bounded by the $\sum_{|\alpha| \le 1} \|D^\alpha \widetilde{\eta}_v\|_{L_2}$, which is polynomial with $1/v$. Furthermore, it can be checked that $\mu$ converges to zero polynomially with $v \to 0$. Under Tsybakov's noise condition, the convergence rate can be obtained via the proof of Theorem 3.1 as

$$L(\widehat{f}) - L^* = O_{\mathbb{P}}(C^{\frac{1}{\kappa+2}} n^{-\frac{d(\kappa+1)}{(2d-1)(\kappa+2)}}) = O_{\mathbb{P}}\left(\text{poly}\left(\frac{1}{v}\right) C^{\frac{1}{\kappa+2}} n^{-\frac{d(\kappa+1)}{(2d-1)(\kappa+2)}}\right).$$

In the Gaussian noise injection case, if we choose $v = v_n = n^{-1/2}$, applying Lemma 3.5 yields

$$L(\widehat{f}) - L^* = O_{\mathbb{P}}(e^{-n\gamma/6}\text{poly}(n)) = O_{\mathbb{P}}(e^{-n\gamma/7}).$$

### E.7 Proof of Theorem 3.7

Direct computation implies that

$$f_i^*(\boldsymbol{x}) = \sum_{j=1}^K \eta_j(\boldsymbol{x}) v_{ji},$$

which implies

$$f^*(\boldsymbol{x}) = (\boldsymbol{v}_1, ..., \boldsymbol{v}_K)\eta(\boldsymbol{x}), \tag{E.27}$$

where $v_{ji}$ is the $i$-th element of $\boldsymbol{v}_j$. Let $\boldsymbol{V} = (\boldsymbol{v}_1, ..., \boldsymbol{v}_K)$. Multiplying $\boldsymbol{V}^\top$ on both sides of Equation E.27 leads to

$$\boldsymbol{V}^\top f^*(\boldsymbol{x}) = \boldsymbol{V}^\top (\boldsymbol{v}_1, ..., \boldsymbol{v}_K)\eta(\boldsymbol{x}) = \left(\frac{K}{K-1}\boldsymbol{I} - \frac{1}{K-1}\boldsymbol{1}\boldsymbol{1}^\top\right)\eta(\boldsymbol{x})$$
$$= \frac{K}{K-1}\eta(\boldsymbol{x}) - \frac{1}{K-1}\boldsymbol{1}\boldsymbol{1}^\top \eta(\boldsymbol{x})$$
$$= \frac{K}{K-1}\eta(\boldsymbol{x}) - \frac{1}{K-1}\boldsymbol{1}, \tag{E.28}$$

where $\boldsymbol{1} = (1, ..., 1)^\top$. In Equation E.28, the second equality is because $\boldsymbol{v}_i^\top \boldsymbol{v}_j = -1/(K-1)$ if $i \ne j$ and $\boldsymbol{v}_i^\top \boldsymbol{v}_i = 1$; and the last equality is because $\sum_{i=1}^n \eta_i(\boldsymbol{x}) = 1$. By Equation E.28, it can be seen that

$$\eta_j(\boldsymbol{x}) = \frac{(K-1)\boldsymbol{v}_j^\top f^*(\boldsymbol{x}) + 1}{K},$$

which finishes the proof.

# F Proof of Lemmas in the Appendix

## F.1 Proof of Lemma E.2

We follow the approach in the proof of Lemma 6.1 and Proposition 6.3 in [23]. Note that $f_{l,P} = 2p - 1$ minimizes $\mathcal{R}_{l,P}$. We first show that for all $f \in \mathcal{F}$ and all $\alpha \geq 0$,

$$\mathbb{E}_{X,Y \sim P}(l_1 \circ f - l_1 \circ f_{l,P})^2 \leq C_{\eta,\kappa}(\|f\|_\infty + 1)^{\frac{2\kappa + 4\alpha}{\kappa + \alpha}} \left\|(2\eta - 1)^{-1}\right\|_{q,\infty}^{\frac{\alpha}{\kappa + \alpha}} \mathbb{E}_{X,Y \sim P}(l_1 \circ f - l_1 \circ f_{l,P})^{\frac{\kappa}{\kappa + \alpha}}, \tag{F.1}$$

where $C_{\eta,\kappa} := \left\|(2\eta - 1)^{-1}\right\|_{\kappa,\infty} + 4$. In particular, one can take $\alpha = 0$ and obtain

$$\mathbb{E}_{X,Y \sim P}(l_1 \circ f - l_1 \circ f_{l,P})^2 \leq C_{\eta,\kappa}(\|f\|_\infty + 1)^2 \mathbb{E}_{X,Y \sim P}(l_1 \circ f - l_1 \circ f_{l,P}). \tag{F.2}$$

Clearly, Tsybakov's noise condition implies that $\left\|(2\eta - 1)^{-1}\right\|_{\kappa,\infty}$ exists. For $x \in \Omega$, let $p := \mathbb{P}(Y = 1|x)$ and $t := f(x)$. Without loss of generality, let $p > 1/2$. Additionally, we denote

$$\begin{aligned}
v(p,t) &= p\left(l(1,t) - l\left(1, f_{l,P}(x)\right)\right)^2 + (1-p)\left(l(-1,t) - l\left(-1, f_{l,P}(x)\right)\right)^2, \\
m(p,t) &= p\left(l(1,t) - l\left(1, f_{l,P}(x)\right)\right) + (1-p)\left(l(-1,t) - l\left(-1, f_{l,P}(x)\right)\right). 
\end{aligned} \tag{F.3}$$

Note $f_{l,P} = 2p - 1$ implies $l\left(1, f_{l,P}(x)\right) = 4(p-1)^2$ and $l\left(-1, f_{l,P}(x)\right) = 4p^2$. Plugging them into Equation F.3, it can be checked that

$$\begin{aligned}
m(p,t) &= 1 + t^2 + 2(1-2p)t - 4p(1-p) = (1 + t - 2p)^2, \\
v(p,t) &= (1 + t - 2p)^2((t+1)^2 + 12p - 4pt - 12p^2).
\end{aligned}$$

By taking

$$\alpha \geq \frac{\log 4 - \log(12p - 12p^2 - 4pt + 2 - (t-1)^2)}{\log|2p - 1|}, \tag{F.4}$$

it can be shown that

$$v(p,t) \leq \left(2t^2 + \frac{4}{|2p-1|^\alpha}\right) m(p,t). \tag{F.5}$$

Since

$$\frac{\log 4 - \log(12p - 12p^2 - 4pt + 2 - (t-1)^2)}{\log|2p-1|} \leq \frac{\log 4 - \log(-\frac{2}{3}t^2 + 4)}{\log|2p-1|} \leq 0,$$

it suffices to take $\alpha \geq 0$. We further define

$$\begin{aligned}
g(y, x) &:= l(y, f(x)) - l(y, f_{l,P}(x)), \\
h_1(x) &:= \eta(x)g(1,x) + (1 - \eta(x))g(-1,x), \\
h_2(x) &:= \eta(x)g^2(1,x) + (1 - \eta(x))g^2(-1,x).
\end{aligned}$$

Therefore, Equation F.5 implies $h_2(x) \leq (2\|f\|_\infty^2 + \frac{4}{|2\eta(x)-1|^\alpha})h_1(x)$ for all $x$ with $\eta(x) \neq 1/2$. Hence, we obtain

$$\begin{aligned}
\mathbb{E}_{X,Y \sim P} g^2 &= \int_{\{x \mid |2\eta(x)-1|^{-1} < t\}} h_2(x) dP_X + \int_{\{x \mid \infty > |2\eta(x)-1|^{-1} \geq t\}} h_2(x) dP_X \\
&\leq (2\|f\|_\infty^2 + 4t^\alpha) \int_{\{x \mid |2\eta(x)-1|^{-1} < t\}} h_1(x) dP_X + \int_{\{x \mid \infty > |2\eta(x)-1|^{-1} \geq t\}} (\|f\|_\infty + 1)^4 dP_X \\
&\leq 4(\|f\|_\infty^2 + t^\alpha)\mathbb{E}_{X,Y \sim P} g + (\|f\|_\infty + 1)^4 \left\|(2\eta - 1)^{-1}\right\|_{q,\infty} t^{-\kappa} \\
&\leq 4t^\alpha(\|f\|_\infty + 1)^2 \mathbb{E}_{X,Y \sim P} g + (\|f\|_\infty + 1)^4 \left\|(2\eta - 1)^{-1}\right\|_{q,\infty} t^{-\kappa} \\
&\leq 4t^\alpha(\|f\|_\infty + 1)^2 \mathbb{E}_{X,Y \sim P} g + (\|f\|_\infty + 1)^4 \left\|(2\eta - 1)^{-1}\right\|_{\kappa,\infty} t^{-\kappa} \\
&\leq C_{\eta,\kappa}(\|f\|_\infty + 1)^{\frac{2\kappa + 4\alpha}{\kappa + \alpha}} \left\|(2\eta - 1)^{-1}\right\|_{\kappa,\infty}^{\frac{\alpha}{\kappa + \alpha}} \mathbb{E}_{X,Y \sim P} g^{\frac{\kappa}{\kappa + \alpha}},
\end{aligned}$$

where the last equality is implied by taking $t^{\kappa+\alpha} := (||f||_\infty + 1)^2(\mathbb{E}_{X,Y\sim P}g)^{-1}$. This shows Equation F.1 holds.

Based on Equation F.1, we can show that Lemma E.1 holds. To see this, let $\widehat{C} := (K\gamma + 1)^{(2\kappa+4\alpha)/(\kappa+\alpha)}$ and fix an $f \in \gamma B_{\mathcal{N}}$. The term $\mathbb{E}_{X,Y\sim P}(L_1 \circ f - L_1 \circ f_n)^2$ can be bounded by

$$
\begin{aligned}
&\mathbb{E}_{X,Y\sim P}(L_1 \circ f - L_1 \circ f_n)^2\\
\leq &2\mu^2 n^{-2}||f||^4 + 2\mu^2 n^{-2}||f_n||^4 + 2\mathbb{E}_{X,Y\sim P}(l_1 \circ f - l_1 \circ f_n)^2\\
\leq &4\mathbb{E}_{X,Y\sim P}(l_1 \circ f - l_1 \circ f_{l,P})^2 + 4\mathbb{E}_{X,Y\sim P}(l_1 \circ f_{l,P} - l_1 \circ f_n)^2 + 2\mu^2 n^{-2}||f||^4 + 2\mu^2 n^{-2}||f_n||\\
\leq &8C_{\eta,\kappa}\widehat{C}(\mathbb{E}_{X,Y\sim P}(l_1 \circ f - l_1 \circ f_{l,P}) + \mathbb{E}_{X,Y\sim P}(l \circ f_n - l \circ f_{l,P}))^{\kappa/(\kappa+\alpha)} + 2\mu^2 n^{-2}||f||^4 + 2\mu^2||f_n||^4\\
\leq &C\widehat{C}(\mathbb{E}_{X,Y\sim P}(l \circ f - l \circ f_{l,P}) + \mathbb{E}_{X,Y\sim P}(l \circ f_n - l \circ f_{l,P}) + \mu^2 n^{-2}||f||^4 + \mu^2 n^{-2}||f_n||^4)^{\kappa/(\kappa+\alpha)}\\
\leq &C\widehat{C}(\mathbb{E}_{X,Y\sim P}(L_1 \circ f - L_1 \circ f_n) + 2\mathbb{E}_{X,Y\sim P}(l \circ f_n - l \circ f_{l,P}) + 2\mu n^{-1}||f_n||^2)^{\kappa/(\kappa+\alpha)}\\
\leq &C\widehat{C}(\mathbb{E}_{X,Y\sim P}(L_1 \circ f - L_1 \circ f_n))^{\kappa/(\kappa+\alpha)} + 2C\widehat{C}a^{\kappa/(\kappa+\alpha)}(\mu).
\end{aligned}
\tag{F.6}
$$

In Equation F.6 the first and second inequalities is because of the Cauchy-Schwarz inequality; the third inequality is because of Equation F.1 and $a^p + b^p < 2(a+b)^p$ for all $a, b \geq 0, 0 < p \leq 1$; the fourth inequality follows from $a^p + b^p < 2(a+b)^p$ for all $a, b \geq 0, 0 < p \leq 1$; the fifth inequality is because $n^{-1}\mu||f||^2 \leq 1$ and $n^{-1}\mu||f_n||^2 \leq 1$; the last inequality follows $(a+b)^p < a^p + b^p$ for all $a, b \geq 0, 0 < p \leq 1$. This finishes the proof of Lemma E.1.

### F.2 Proof of Lemma E.4

Since there is a positive margin between $\Omega_1$ and $\Omega_2$, we can always find two sets $\widetilde{\Omega}_1$ and $\widetilde{\Omega}_2$ with infinitely smooth boundaries such that $\Omega_1 \subset \widetilde{\Omega}_1$, and $\Omega_2 \subset \widetilde{\Omega}_2$. Then the result follows from the Sobolev extension theorem.

## G Appendix for Detailed Experiments

### G.1 Synthetic Data

We consider the square loss based and cross-entropy based overparameterized neural networks (ONN) with $\ell_2$ regularization, denoted as SL-ONN + $\ell_2$ and CE-ONN + $\ell_2$, respectively. The chosen ONNs are two-hidden-layer ReLU neural networks with 500 neurons for each layer, and the parameter $\mu$ is selected via a validation set. During the neural network training, we use the popular RMSProp optimizer with the default settings, and select the tuning parameter $\mu$ for SL-ONN + $\ell_2$ and CE-ONN + $\ell_2$ by a validation set.

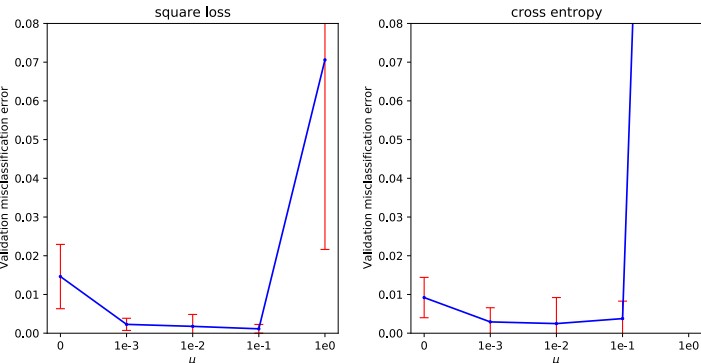

Figure G.3: The errorbar plot of validation misclassification rate with respect to different $\mu$ in the separable case.

**Separable case**  In the separable case, we consider a two-dimension distribution $P = (\rho \sin\theta + 0.04, \rho \cos\theta)$ where $\rho = (\theta/4\pi)^{4/5} + \epsilon$ with selected $\theta$ from $(0, 4\pi]$ and $\epsilon \sim \text{unif}([-0.03, 0.03])$. We draw 100 positive and 100 negative training samples from $-P$ and $P$, respectively. We select the tuning parameter $\mu$ for SL-ONN + $\ell_2$ and CE-ONN + $\ell_2$ by minimizing the validation misclassification rate, where the candidate set of $\mu$ is $\{0, 0.001, 0.01, 0.1, 1\}$. For each $\mu$, we generate 40 replications to estimate the mean and standard deviation of validation misclassification rate. We observe that SL-ONN + $\ell_2$ and CE-ONN + $\ell_2$ have the least mean and least standard deviation for the validation misclassification rate at $\mu = 0.1$ and $\mu = 0.01$, respectively. The errorbar plot for each $\mu$ is shown in Figure G.3.

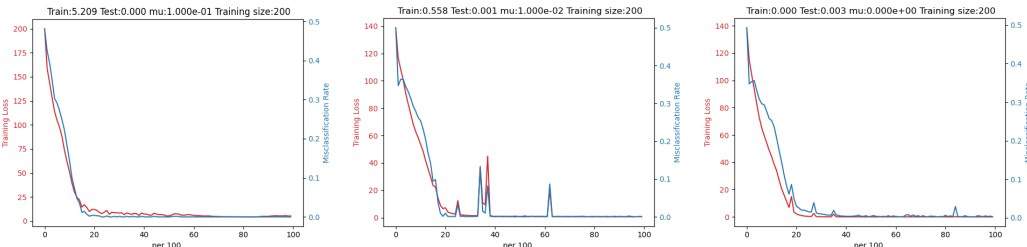

Figure G.4: An instance about the training process of SL-ONN + $\ell_2$ *(Left)*, CE-ONN + $\ell_2$ *(Center)* and CE-ONN *(Right)* in the separable case.

We also consider the cross-entropy loss based ONN without $\ell_2$ regularization (CE-ONN). All three models are trained for 10000 iterations and achieve 0% training misclassification rate. In Figure G.5, we present five more examples about the decision boundary prediction and test accuracy of SL-ONN + $\ell_2$, CE-ONN + $\ell_2$ and CE-ONN. We can find that SL-ONN + $\ell_2$ still beats CE-ONN + $\ell_2$ and CE-ONN in almost all the cases. SL-ONN + $\ell_2$ attains the smallest misclassification rate and depicts the largest margin decision boundary which separates the positive and negative samples best. In addition, we can observe that CE-ONN + $\ell_2$ outperforms CE-ONN in all cases, although the $\ell_2$ regularization term bring some oscillations to the training of CE-ONN + $\ell_2$, as shown in Figure G.4.

For providing concrete numerical verification, we also conduct an experiment to compare the margins of decision boundaries trained by the square loss, as shown in Figure G.7. Moreover, Figure G.7 also presents the margin of the decision boundary predicted by the cross entropy with adversarial training, where we adapt the PGD attack with the attack strength level 0.05. It is obvious that the margins of square loss is larger than that of cross entropy, and is comparable to that of cross entropy with adversarial training. Meanwhile, the decision boundary of cross entropy with adversarial training is more smooth than that of cross entropy. We note that hereby the decision boundary of cross entropy with adversarial training can be regarded as the optimal decision boundary. Hence, Figure G.7 verifies the robustness of the square loss and demonstrates that the decision boundary predicted by square loss is closer to the optimal one.

**Non-separable case**  We consider the conditional probability $\eta(\boldsymbol{x}) = \sin(\sqrt{2}\pi\|\boldsymbol{x}\|_2), \boldsymbol{x} \in [-1, 1]^2$, and the calibration performance of SL-ONN + $\ell_2$ and CE-ONN + $\ell_2$, where the classifiers are denoted by $\widehat{f}_{l2}$ and $\widehat{f}_{ce}$, respectively. The training data points $\boldsymbol{x}_1, \cdots, \boldsymbol{x}_n$ are i.i.d. sampled from $\text{unif}([-1, 1]^2)$ and the training labels $y_1, \cdots, y_n$ are sampled according to $Bernoulli(\eta(\boldsymbol{x}_i))$, where $\eta(\boldsymbol{x}) = \sin(\sqrt{2}\pi\|\boldsymbol{x}\|_2)$, and $n = 8000$. The 3-dimensional plot of $\eta(\boldsymbol{x})$ is presented in Figure G.6.

We select the tuning parameter $\mu$ for SL-ONN + $\ell_2$ and CE-ONN + $\ell_2$ via a validation set, and the candidate set of $\mu$ is $\{0, 0.001, 0.01, 0.1, 1\}$. For each $\mu$, we run 40 replications to estimate the mean and standard deviation of validation misclassification rate. The iteration number of training is 2000. We find SL-ONN + $\ell_2$ and CE-ONN + $\ell_2$ have the smallest mean and standard deviation for the validation misclassification rate at $\mu = 0.1$ and $\mu = 0.1$, respectively. The error bar plot [5] for $\mu$ equaling to 0, 0.001, 0.01, 0.1 and 1 is shown in Figure G.8.

The calibration error results are presented in Figure G.9. The error bar plot of the test calibration error shows that $\widehat{f}_{l2}$ has the smaller mean and standard deviation than $\widehat{f}_{ce}$. It suggests that square

---
[5]In an error bar plot, the center of each plot is the mean, and the upper and lower red dashes denote (mean+one standard deviation) and (mean − one standard deviation), respectively.

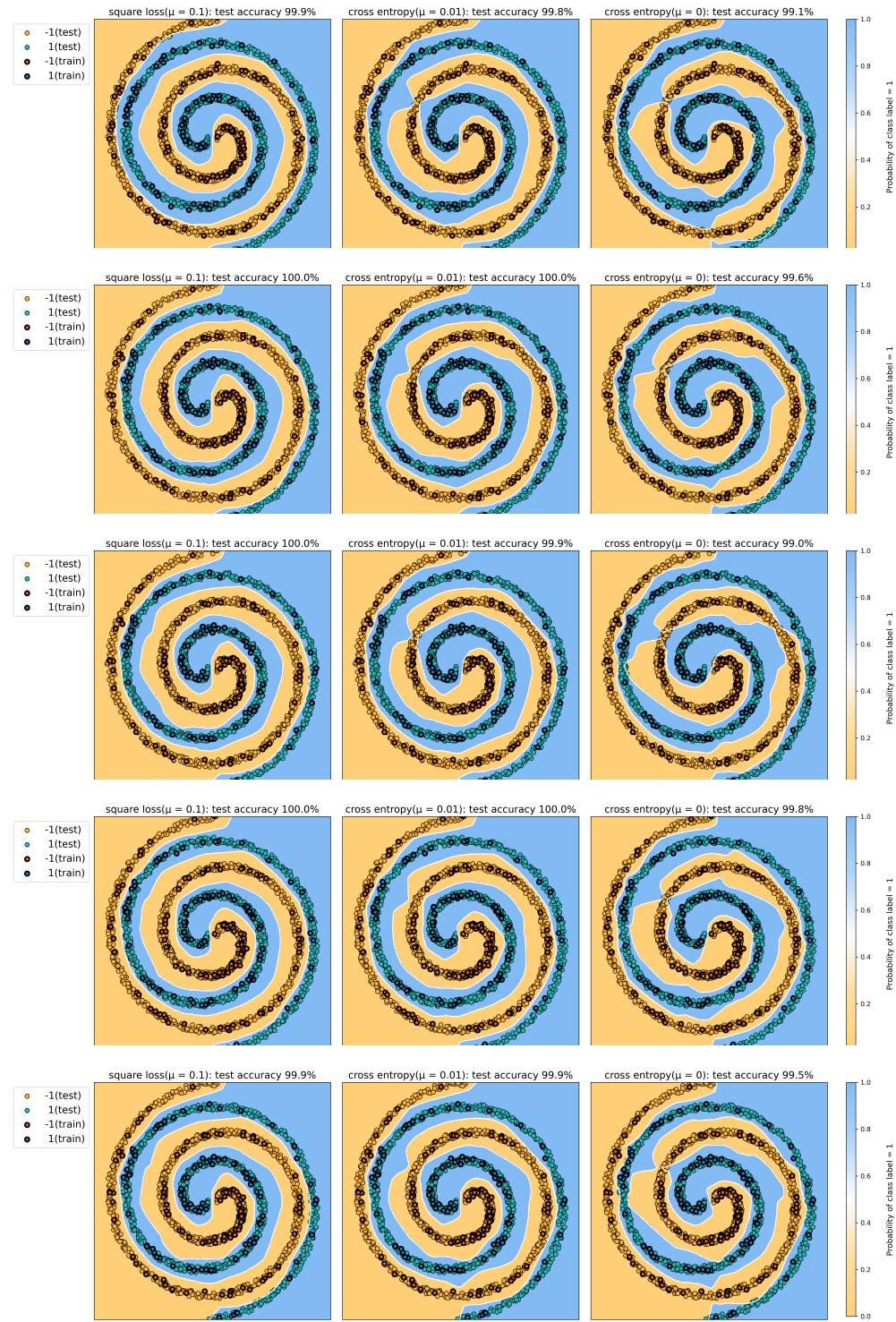

Figure G.5: Five examples in the separable case.

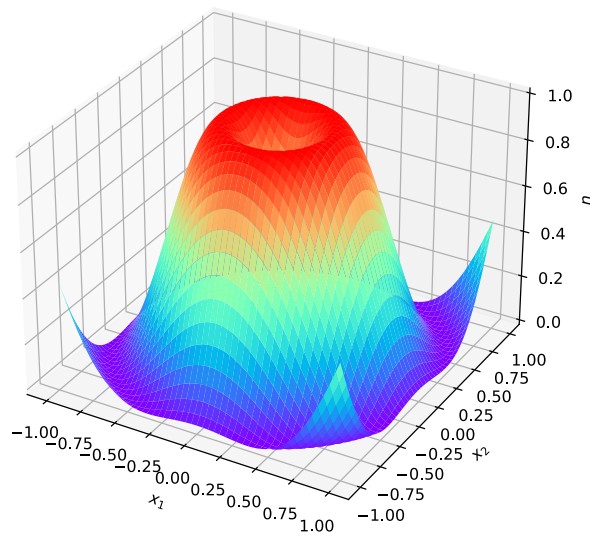

Figure G.6: The 3-dimensional plot of $\eta(\boldsymbol{x})$ in the non-separable case.

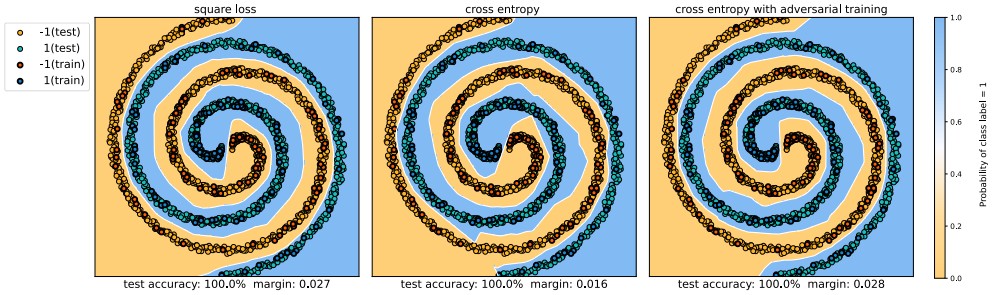

Figure G.7: Test misclassification rates, decision boundaries and margins predicted by: SL-ONN + $\ell_2$ (Left); CE-ONN + $\ell_2$ (Center); CE-ONN with adversarial training in the separable case.

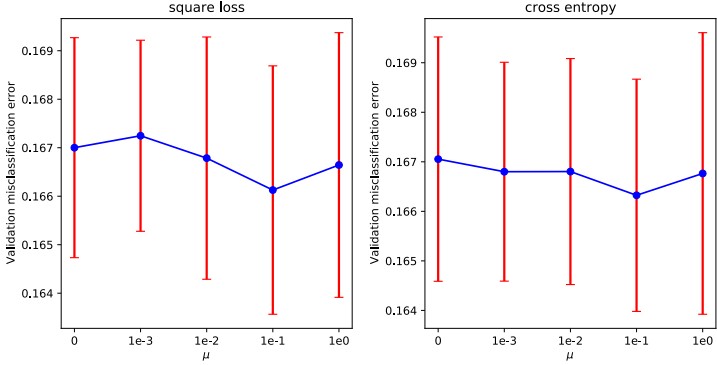

Figure G.8: The errorbar plot of validation misclassification rate with respect to different $\mu$ in the non-separable case.

loss generally outperforms cross entropy in calibration. The histogram and kernel density estimation of the test calibration errors for one case show that the pointwise calibration errors on the test points of $\widehat{f}_{l2}$ are more concentrated around zero than those of $\widehat{f}_{ce}$. Moreover, despite a comparable misclassification rate with $\widehat{f}_{ce}$, $\widehat{f}_{l2}$ has a smaller calibration error. Figure G.9 demonstrates that SL-ONN + $\ell_2$ recovers $\eta$ much better than CE-ONN + $\ell_2$.

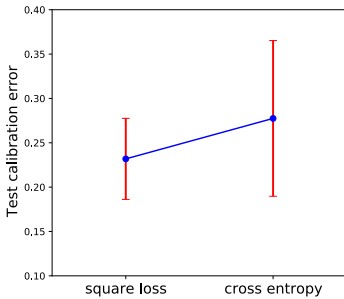 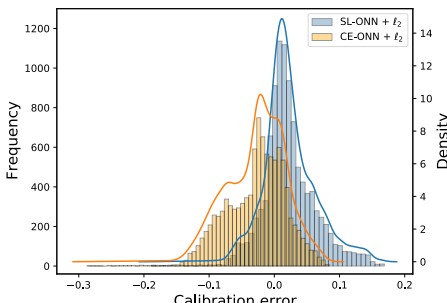

Figure G.9: (Left) The error bar plot of test calibration errors for the 40 replicated runs. (Right) The histogram and kernel density estimation of test calibration errors in one instance from 40 replications. In this instance, $||(\widehat{f}_{l2} + 1)/2 - \eta||_{L_\infty} = 0.188$, $||(\widehat{f}_{ce} + 1)/2 - \eta||_{L_\infty} = 0.287$, and the test misclassification rate for $\widehat{f}_{l2}$ and $\widehat{f}_{ce}$ are 0.167 and 0.164, respectively.

## G.2   Real Data

**Data and network architecture**   We use the popular CIFAR-10 and CIFAR-100 datasets, with training and testing split of 50000 and 10000. The data loader is from `torch.utils.data`. As typically employed in practice, our training includes data augmentations, a composition of random crop and horizontal flip. We trained two types of neural networks, ResNet [50] and Wide ResNet [51]. To be more specific, we used the default ResNet-18 and ResNet-50 for CIFAR-10 and CIFAR-100 respectively, and the default WRN-16-10 for both CIFAR-10 and CIFAR-100. All experiments are run in PyTorch version 1.9.0 and cuda 10.2.

**Training details**   The training algorithm is the default SGD with momentum (0.9) and weight decay (0.0005). The learning rate scheduler is the `StepLR()` from `torch.optim.lr_scheduler` with step size 50. In our experiment, the only parameters that we tuned are the learning rate (lr) and batch size (bs), with only two options, (lr=0.01, bs=32) and (lr=0.1, bs=128). We find that (lr=0.01, bs=32) performs better for most cases except for square loss trained WRN-16-10 on CIFAR-100, where the average accuracy for (lr=0.01, bs=32) is 77.96%, around 1.5% less than that for (lr=0.1, bs=128). Meanwhile, for cross-entropy trained WRN-16-10, (lr=0.1, bs=128) yields an average accuracy of 76.83%, around 1% less than that for (lr=0.01, bs=32). The two training settings perform quite comparable for WRN-16-10 on CIFAR-10. For consistency, we stick with (lr=0.01, bs=32) in this case.

**Adversarial robustness**   For square loss, training deep classifiers is the same as regression. When attacking classifiers trained with square loss, the default way of constructing adversarial examples doesn't work well. To be more specific, for a correctly classified training image $(\boldsymbol{x}, y)$, the adversarial examples are typically generated by

$$\max_{\|\boldsymbol{\delta}\|_\infty = \alpha} L(f(\boldsymbol{x} + \boldsymbol{\delta}), y).$$

Such an attacking scheme works fine for cross-entropy, where

$$L(f(\boldsymbol{x}), y) = -\log(\text{softmax}(f(\boldsymbol{x}))) = -\log\left(\frac{\exp(f_y(\boldsymbol{x}))}{\sum_{k \neq y} \exp(f_k(\boldsymbol{x}))}\right),$$

but is problematic for regression losses such as square loss. The fundamental reason lies in Proposition 3.7 and its proof. Recall that the conditional probability for square loss consists of projections of

the classifier outputs to all the simplex vertices, some of which are sure to be non-positive. The sum of the class probabilities from Equation 3.2 is always 1 but unlike that from softmax function, the summand can be negative. By maximizing the square loss, the resulting "adversarial" image can stay the same class but more confidently. To illustrate, if $f(\boldsymbol{x}) = \boldsymbol{v}_y$, the predicted confidence for label $y$ will be 100% and 0 for other classes. The "adversarial" image may be such that $f(\boldsymbol{x} + \delta) = 2\boldsymbol{v}_y$, where the predicted label remains unchanged but with an updated confidence of $2 - 1/K$ for label $y$ and $(1/K - 1)/(K - 1) < 0$ for all other classes. This is obviously not a successful attack.

To this end, we devise a special attacking scheme for classifier trained with square loss and simplex coding. The key idea is to choose attack directions tangent to the sphere inscribed by the simplex. Instead of

$$L(f(\boldsymbol{x}), y) = \|f(\boldsymbol{x}) - \boldsymbol{v}_y\|_2^2,$$

we choose

$$L(f(\boldsymbol{x}), y) = \theta(f(\boldsymbol{x}), \boldsymbol{v}_y),$$

where $\theta(\boldsymbol{v}_1, \boldsymbol{v}_2)$ denotes the cosine similarity between $\boldsymbol{v}_1$ and $\boldsymbol{v}_2$. We refer to this attack as angle attack.

Empirically, we found our angle attack to significantly outperform the naive attack by maximizing the square loss. For square loss, let the predicted probabilities from Equation 3.2 be $\widehat{\boldsymbol{p}}$. Similar to cross entropy, we have also tried two cases of $L(f(\boldsymbol{x}), y)$, which corresponds to

$$L_1(f(\boldsymbol{x}), y) = -\log(\mathrm{softmax}(\widehat{p}_y(\boldsymbol{x}))) \quad \text{and} \quad L_2(f(\boldsymbol{x}), y) = -\log(\widehat{p}_y(\boldsymbol{x})).$$

Interestingly for PGD-100, $L_1$ performs the best, beating angle attack for the majority cases, except for attacking WRN-16-10 on CIFAR-100 with strength 2/255. The reported adversarial accuracy for square loss trained classifiers in Table 1 is by $L_1(f(\boldsymbol{x}), y) = -\log(\mathrm{softmax}(\widehat{p}_y(\boldsymbol{x})))$.

The PGD attack results may be further improved for square loss. Nonetheless, the AutoAttack still provides convincing results, as it includes both white-box and black-box attacks. We used the standard version which includes 4 types of attacks, APGD-CE, APGD-DLR, FAB and Square Attack as in [54].

**Robustness to Gaussian Noise**    To make the robustness evaluation more comprehensive, beyond the adversarial robustness, we also investigate the classifier's robustness to Gaussian noise injections. With the image pixels' value normalized to 0 and 1, we consider injecting Gaussian noises to test images and report the test accuracy. The noise standard deviation ranges from 0.1 to 0.4. The test accuracy results for both CIFAR-10 and CIFAR-100 are listed in Table G.3.

Table G.3: Black-box Gaussian noise robustness results. The reported accuracy is the average of 5 replications.

| Dataset | Network | Loss | Gaussian noise standard deviation | | | | |
|---|---|---|---|---|---|---|---|
| | | | 0.00 | 0.10 | 0.20 | 0.30 | 0.40 |
| CIFAR-10 | ResNet-18 | SL | 95.04 | **90.07** | **70.16** | **42.13** | **25.38** |
| | | CE | **95.15** | 90.03 | 69.71 | 41.08 | 24.66 |
| | WRN-16-10 | SL | **95.02** | **88.49** | **60.91** | **35.78** | **24.04** |
| | | CE | 93.94 | 84.78 | 56.63 | 33.70 | 22.41 |
| CIFAR-100 | ResNet-50 | SL | **78.91** | **63.06** | **36.64** | **17.78** | **9.47** |
| | | CE | 79.82 | 62.72 | 34.42 | 16.69 | 9.11 |
| | WRN-16-10 | SL | **79.65** | **62.01** | **30.69** | **15.11** | **8.88** |
| | | CE | 77.89 | 60.14 | 26.47 | 10.26 | 5.57 |

**Simplex coding vs. one-hot coding**    The one-hot coding is the usual choice for applying square loss to classification. However, it is empirically observed to struggle when the number of classes are large. For a single training data point $\boldsymbol{x}$ and label $k$, [4] proposed to modify the training objective from the typical $(f_k(\boldsymbol{x}) - 1)^2 + \sum_{i \neq k} f_i(\boldsymbol{x})^2$ to $J \cdot (f_k(\boldsymbol{x}) - M)^2 + \sum_{i \neq k} f_i(\boldsymbol{x})^2$, where $J, M$ are hyperparameters to make $f_k$ more prominent in the loss. Similar modification is also proposed in [7]. The scaling trick involves two hyperparameters, which can be hard to tune. We evaluate the two coding schemes in our experiment setting and the results are summarized in Table G.4. The test accuracy for scaled one-hot coding scheme performs comparably for ResNet-18 on CIFAR-10 and ResNet-50 on CIFAR-100. For WRN-16-10, the simplex coding performs better.

Table G.4: Test accuracy for square loss with one-hot coding (scaled) (OC) vs. simplex coding (SC). Accuracy with an asteroid sign ($^*$) denotes cases where the training accuracy doesn't overfit after 200 training epochs.

| Dataset | Network | One-hot scaling | SGD parameters | OC clean acc(%) | SC clean acc(%) |
|---------|---------|-----------------|----------------|-----------------|-----------------|
| CIFAR-10 | ResNet-18 | k=1, M=1 | lr=0.01, bs=32 | 94.95 | 95.04 |
| | | | lr=0.1, bs=128 | 10$^*$ | 10$^*$ |
| | WRN-16-10 | k=1,M =1 | lr=0.01, bs=32 | 89.75$^*$ | 95.02 |
| | | | lr=0.1, bs=128 | 88.43$^*$ | 95.03 |
| CIFAR-100 | ResNet-50 | k=5, M=15 | lr=0.01, bs=32 | 79.06 | 78.91 |
| | | | lr=0.1, bs=128 | 1$^*$ | 1$^*$ |
| | WRN-16-10 | k=5, M=15 | lr=0.01, bs=32 | 78.42 | 78.06 |
| | | | lr=0.1, bs=128 | 78.39 | 79.65 |