# OpenReview forum: "Understanding Square Loss in Training Overparametrized Neural Network Classifiers"
_NeurIPS.cc/2022/Conference — NeurIPS 2022 Accept_

### Official Review · Reviewer_bBhR · 2022-07-07

**Rating:** 8
**Confidence:** 4
**Soundness:** 4 excellent
**Presentation:** 4 excellent
**Contribution:** 3 good

**Summary:**

This paper provides solid arguments in favor of using MSE loss instead of the most common CE loss. From several aspects, showing that:
- MSE is better calibrated.
- It has a better generalization bound.
- Is more robust to adversarial perturbations.

**Questions:**

- CE is shown to converge to the max-margin solution. So I don't fully understand Thm 3.3. Yes, MSE's margin is bounded from below but is still worse to equal to margin of CE. I am right?
- I do not fully understand Fig. 2. How do you measure the confidence? The reason I'm asking this is because the output with MSE is real valued and not a probability. No?

**Strengths And Weaknesses:**

** Strengths**:
- I loved the structure of this work. Well-motivated, properly stated, and provide a good balance between the theory and intuition.
- I found the results to be novel and significant.
- The experiments appear sufficient to support the claims.
- The authors are fair in their evaluations. For example, I enjoyed reading your argument on why white-box attacks appear to fail with MSE.

** Weaknesses**:
- There are a few citations missing. I know at least the following three which advocate using MSE instead of CE:
    + "Excessive invariance causes adversarial vulnerability" by Jacobsen et al
    + "Cross-entropy loss and low-rank features have responsibility for adversarial examples" by Nar et al.
    + "Gradient starvation: A learning proclivity in neural networks" by Pezeshki et al.
- The writing can benefit from a grammar check. For example, "it's" --> "it is".
- The use of separable / non-separable terminology was initially very confusing for me. Generally "separable" refers to a case that data is linearly separable or the model has enough capacity to fully classify the data. However, here, authors refer to label ambiguity.
- The extension to multi-class using simplex coding is rather difficult to follow.
- The code for the experiments is not provided.

---

> ### Author Response · Authors · 2022-08-02
> **Response to Reviewer bBhR**
>
> ## Response to Weakness
>
> 1. Thanks for pointing out the reference and we have added them in the related work.
>
> 2. Thank you. We have checked the paper and made corrections.
>
> 3. We added more explanation around the separable / non-separable terminology to make it clear.
>
> 4. We have re-organized the text of Section 3.4
>
> 5. This work mainly contributes to the theoretical understanding of using square loss for neural network based classification. Numerical experiments are conducted to corroborate our theoretical results, rather than establishing new state-of-the-art benchmarks.
> Hence, our experiments only considered fully connected ReLU networks on simulated data and (Wide) ResNet on the CIFAR dataset, based on standard benchmark settings. We did not propose any new DNN classification algorithm in this work. In our experiment, the modification from CE to SL with simplex coding is rather straightforward. Though we did not provide the code, we have listed the training setup details and the hyperparameter choices in the Appendix.
>
> ## Response to Q1
> Thank you for your question. We think the paper you refer to is "Gradient descent maximizes the margin of homogeneous neural networks". You are correct that the max-margin solution may be more desired. However, it should be noted that in that paper,
> the margin refers to the distance between the estimated decision boundary with the **training** samples, while in our work, we show that the margin of the **true** underlying classes has a lower bound.  We intend to study whether SL can also converge to the max-margin solution with respect to the training samples in the future.
>
> Also, empirical and simulation results indicate that SL tends to have a larger margin than CE in practice. For more visualization, we added a new Figure G.7  for the seperable case in $[-1,1]^2$ in the Appendix, which not only illustrates the decision boundaries predicted by square loss, cross entropy, and cross entropy with adversarial training but also their corresponding $l_2$-margins for the **test data**.
>  The results of margins are also summarized in the following table.
> |       | square loss |cross entropy | cross entropy with adversarial training |
> | :----:| :----: | :----:|:----:|
> | $l_2$-margin | 0.027 | 0.015 | 0.028|
>
> As we can see, the margin of square loss is larger than that of cross entropy and close to that of cross entropy with adversarial training.
> We note that hereby the decision boundary predicted by cross entropy with adversarial training can be regarded as the optimal one.
> Hence, the above table verifies the robustness of square loss and the decision boundary predicted by square loss is much closer to the optimal one.
>
> ## Response to Q2
> Thanks for the question.
> We should clarify that the optimal confidence is given by the ground truth conditional probability $P(y=1|x)$.
> Like CE, SL can also give confidence in classification.
> The 2-class case (with label -1, 1) is simpler where we can get the estimated confidence as $(f(x)+1)/2$, as stated in Theorem 3.4. Our theorem 3.4 gives a theoretical guarantee on the calibration error under SL.
>
> For multi-class classification using SL, the confidence can be tricky. In Proposition 3.7, we gave the formula, which depends on projections to each of the label coding. The proof can be found in Appendix E.7.
>
> You are correct that the estimated confidence/conditional probability can be larger than 1 or negative.
> However, if the SL is properly minimized, our Theorem 3.4 states that the predicted confidence will be close to the ground truth. More discussion on this property and how it is related to constructing adversarial examples can be found in Appendix G.2.

---

> > ### Comment · Reviewer_bBhR · 2022-08-03
> > **Thank you for the reply!**
> >
> > I appreciate that authors' reply!
> > Thanks for the clarifications on the margin. As authors correctly pointed out, the margin is wrt the distribution not the training samples. I am just wondering how you compute the l2-margin of a non-linear decision boundary? Do you apply a gradient ascent until you project a point onto the decision boundary?
> >
> > Thanks

---

> > > ### Author Response · Authors · 2022-08-04
> > > **Reply to Reviewer bBhR**
> > >
> > > Thank you for the question!
> > >
> > > For the computation, we did not use gradient ascent, but a less efficient grid search method, since we are considering a very simple toy case.
> > > We divide $[-1, 1]^2$ into 2000 by 2000 grids of equal size. For each grid (not on the boundary), we consider its 8 neighboring grids. If the classifier function gives different signs for them, we identify it as a boundary grid.
> > > After exhaustive calculation, we get a set of points representing the decision boundary.
> > > Then, we calculate the minimum $l_2$ distance of each test point (2000 test points in total) to the boundary points set.
> > > Finally, the margin is computed by taking the minimum of the 2000 $l_2$ distances of test points.
> > >
> > > For more complicated cases in higher dimensions, grid search will be computationally prohibitive. A better way to calculate the margin to nonlinear boundary might be by gradient ascent, as in constructing adversarial attacks. This method has been applied by Ding et al. in
> > > _MMA Training: Direct Input Space Margin Maximization through Adversarial Training_, where they computed the margin by adversarial examples.

---

> > > > ### Comment · Reviewer_bBhR · 2022-08-08
> > > > **Thanks for the clarification**
> > > >
> > > > Thanks for the clarification!

---

### Official Review · Reviewer_rLs9 · 2022-07-07

**Rating:** 5
**Confidence:** 4
**Soundness:** 4 excellent
**Presentation:** 3 good
**Contribution:** 3 good

**Summary:**

The authors explore the use of square loss in classification settings, with both one-hot and scaled simplex features. They motivate the study by invoking improvements in model calibration and feature selection that have been theorized/observed about the use of square loss in classification settings.

The authors then study the generalization error and model calibration in the large width Neural Tangent Kernel (NTK) setting. Under a particular set of conditions on the dataset, they derive non-trivial bounds for the generalization error and the calibration as a function of the number of data points n.

They conclude with experiments showing that using squared loss with scaled simplex features improves performance on CIFAR. They also show that squared loss provides additional robustness to adversarial examples.

**Questions:**

How close do the bounds in equations 3.1 and 3.4 compare to empirical behavior? I would be curious to know how tight they are even in toy examples. I'd also be curious about the relationship between the bounds in this paper and those in http://proceedings.mlr.press/v119/huang20l.html and https://proceedings.mlr.press/v119/adlam20a.html.

What do results of squared loss look like on tasks with a large number of classes, like imagenet or language modeling?

Is there any way to visualize/quantify the difference in decision boundaries between the cross entropy and squared loss cases? This seems particularly interesting in the adversarial setting.

**Limitations:**

Adequately addressed

**Strengths And Weaknesses:**

The analysis of the generalization error and calibration is novel and interesting. I believe that it is correct, though the arguments are long and it is possible I've missed an issue there.

The authors mention that not much work exists in terms of generalization bounds for wide neural networks under squared loss; however, there has been much recent work in this area in the context of the simultaneous scaling of width and data (http://proceedings.mlr.press/v119/huang20l.html, https://proceedings.mlr.press/v119/adlam20a.html). Comparing the author's results to these scenarios would be interesting and useful to position the paper in the literature.

Overall the empirical results seemed the most promising. The fact that both training and adversarial attacks were studied helped solidify the claim that both generalization and calibration can be improved by using square loss. Overall I find the empirical results even more compelling than the theoretical ones due to their careful presentation. I think the experiments could be strengthened by exploring the efficacy of squared loss on datasets with a large number of classes (Imagenet, language modeling), where the SOTA models are far from the NTK regime studied in the theory section of the paper.

---

> ### Author Response · Authors · 2022-08-02
> **Response to Reviewer rLs9**
>
> ## Response to Q1
> Usually, the convergence rate is only theoretical, which depends on many quantities that are not able to identify, e.g., the smoothness of the underlying function, and the constants in the bound. Furthermore, the convergence results are based on the minimax settings, i.e., the **worst case** analysis. Since we cannot cover all functions in numerical studies even in the one-dimensional case, it is typically hard to verify the convergence rate. To the best of the authors' knowledge, we do not know any existing work verifying minimax rate via simulation.
>
> Furthermore, in practice, the performance of a classifier can be influenced by a lot of factors, like neural networks' structures, optimization methods, regularization strength, etc. Therefore, it is even harder to verify the convergence rate for a neural network classifier in practice.
>
>  In this work, we mainly focus on the generalization, robustness, and calibration of the overparameterized neural network classifier. We pay less attention to the training dynamics, which was considered by Huang's work. The bounds in Huang's work are for training dynamics, which are from a different perspective with ours. Although our work relies on training neural networks by gradient descent with weight decay, while Huang's work considered the gradient descent without weight decay, we think Huang's work will be able to relax Assumption D.1 and may generalize our results to multi-layer neural networks, which needs more efforts.
> Nonetheless, this does not significantly affect the message we want to convey.
>
>  The second work is mainly for regression. Furthermore, they considered a specific data distribution and provided an exact analytical characterization of the generalization error. The exact characterization provides new insight that the overparameterization sometimes can lead to triple descent. We consider classification problems, and we focus on the theoretical analysis of SL from different perspectives, including not only generalization performance but also calibration and robustness. However, the high-dimensional settings in the second work are quite inspiring, and we believe it would be interesting to generalize our results to a similar high-dimensional setting. We added a remark in new line 148.
>
> ## Response to Q2
>
> Empirically, [Hui \& Belkin ICLR 2020] showed that SL tends to perform better for NLP tasks, but CE is generally better for CV tasks. We have added a new experiment of ResNet-18 on Tiny-ImageNet, where the findings remain the same, i.e., comparable clean accuracy but better adversarial robustness and calibration. The results are summarized below:
> |  | Clean Acc | AutoAttack($l_\infty $  2/255)  | ECE |
> | :--:| :----: | :----:|:----:|
> | CE | 62.9% | 0. | 0.081|
> | SL | 62.7% | 0.49| 0.059|
>
> The network is ResNet-18. The batch size and learning rate are 128 and 0.2 respectively. The classifier is trained by SGD with weight decay (0.0001).  We run 3 replications and the reported number is the average.
>
> ## Response to Q3
>
> Thanks for your suggestion. For more visualization, we added a new Figure G.7 for the separable case in $[-1,1]^2$ in the Appendix, which not only illustrates the decision boundaries predicted by square loss, cross entropy, and cross entropy with adversarial training but also their corresponding $l_2$-margins for the **test data**.
>  The results of margins are also summarized in the following table.
>
> |       | square loss |cross entropy | cross entropy with adversarial training |
> | :----:| :----: | :----:|:----:|
> | $l_2$-margin | 0.027 | 0.015 | 0.028|
>
> As we can see, the margin of square loss is larger than that of cross entropy and close to that of cross entropy with adversarial training.
> We note that hereby the decision boundary predicted by cross entropy with adversarial training can be regarded as the optimal one.
> Hence, the above table verifies the robustness of square loss and the decision boundary predicted by square loss is much closer to the optimal one.

---

### Official Review · Reviewer_zn8W · 2022-07-10

**Rating:** 5
**Confidence:** 3
**Soundness:** 3 good
**Presentation:** 3 good
**Contribution:** 3 good

**Summary:**

The paper conducted comprehensive review in research, and compare the loss function between Cross Entropy (CE) loss and Square Loss(SL).  The analytical results show that compared to CE, SL has comparable generalization error, but noticeable advantage in robustness and calibration. Experimental results from synthetic data and real data further demonstrate the claim of the paper.

**Questions:**

[1] Theorem 3.1 shows the gap between loss with neural networks and loss with minimize 0-1 loss. As mentioned in equation 2.1, one minimizer is L*., which might not be the optimal minimizer. Could the author help to clarify if L* is bounded with optimal loss with limited error?

[2] From line 62 in page 2, it shows the mis-calibration of CE in prediction region close to 0 or 1 has been shown in paper [9]. Similar results are also covered in Theorem 3.4, between line 248 and 256 in page 6. Could the author help to confirm the originality of the results and if any gaps or difference between this paper and paper [9]?

[3] From the theory analysis, it seems CE tends to weakly calibrated in the high predicted or low predicted region (Theorem 3.4). However, from the experimental results in Fig 2, it seems the mis-calibration happens in low prediction region (confidence < 0.1) and normal prediction region (confidence between 0.5 and 0.8). Does the experimental result align with the theorem?

[4] From the experimental results from Table 1, it seems SL is not always better than CE, although SL seems to be better than CE when under attack. Are there any hypothesis or interpretation to show the situation when CE will outperform than SL?



**Ethics Review Area:**

["I don’t know"]

**Limitations:**

The paper shows the advantage of SL over CE, but not the dis-advantage of SL compared to CE. Might need more insights to compare the disadvantage of SL, and explain the the performance gap between two loss function in experimental results from Table 1.

**Strengths And Weaknesses:**

The paper comprehensive summarized the areas and related results. The paper is well written and clearly organized. The theoretical results of CE calibration issue aligned with the observations and the learning from the theoretical results helps to understand the gaps in practice.

Some hypothesis and learning from the theorem might need more clarification. Experimental results might need further improvement to demonstrate the claim made in the theorem. More details can be found from questions sections.

---

> ### Author Response · Authors · 2022-08-02
> **Response to Reviewer zn8W  (Part 1)**
>
> ## Response to Q1
>
> In Theorem 3.1, $L^*$ is the minimum **population level** 0-1 loss that we can achieve if we know the  **underlying true conditional distribution** $\eta(x)=\mathbb{P}(y=1|x)$. The value of $L^*$, albeit unique, can be achieved by many minimizers. One minimizer, as mentioned in equation 2.1, is $2\eta-1$. See Theorem 2.1 of [9]. In fact, as long as a classifier has the same sign with $2\eta(x)-1$ for all $x\in \Omega$, it is an optimal minimizer and achieves the minimum 0-1 loss $L^*$.
>
>  As for the value of $L^*$ itself, note that by equation 2.1, we have
>
>
>    $L^* = L(2\eta-1)$ = $E_{(X,Y)\sim P}$ $I_{\{{\rm sign}(2\eta(X)-1)\neq Y\}}$
>     $=  \int_{{x: \eta(x)\geq 0.5}}  (1-\eta(x)) {\rm d}P_X + \int_{\{x: \eta(x)< 0.5\}}  \eta(x) {\rm d}P_X,$
>
>
>  which is only related to the underlying true distribution of $(X, Y)$. It can be seen that if $\eta(x)$ is close to 1 or 0 on a large part of $\Omega$, then $L^*$ is small. In particular, in the separable case, $L^*=0$, while in the worst case where $\eta(x)=0.5$ for all $x\in \Omega$, we have $L^*=0.5$, which has the same loss as flipping a coin. Since $L^*$ is only related to the underlying truth, we consider it as a constant and discuss the difference of $L(\hat f)-L^*$ in Theorem 3.1.
>
> ## Response to Q2
> Sorry for the confusion here. In line 62 in page 2 and Theorem 3.4, between lines 248 and 256 in page 6,
>  the optimal solution under CE is not original (classic result).
>  The analysis on calibration error for neural network (line 248) can be directly obtained from the optimal solution form so we did not cite any reference or write it as a formal proposition.
>
>  In this work, the calibration results we derived are specifically for the squared loss. Theorem 3.4 provides a convergence rate of the calibration error for the neural network classifier under square loss, which is not covered by [9].
>  Our work is significantly different from [9] in the following aspects: (1) We considered training overparametrized neural network classifier; (2) We derived explicit excessive risk convergence rate (Theorem 3.1); (3) We derived explicit calibration error convergence rate (Theorem 3.4); (4) We analyzed the separable case and derived margin lower bound (Theorem 3.3).
>  All the aforementioned results are not addressed in [Zhang 2004].
>
> ## Response to Q3
> First, we would like to clarify that Theorem 3.4 only concerns square loss. We did not obtain the $L_\infty$ estimation rate under CE loss and to the best of the authors' knowledge, no such result exists.
> Instead, we provided explanations of how CE can struggle to be well calibrated when the network's capacity is limited.
>
> To provide concrete numerical verification, we conducted 1-d experiment where $P(x|y=1)$ follows Gaussian mixture of $ N(1, \sigma^2)$ and $N(-2, \sigma^2)$, and $P(x|y=-1)$ follows Gaussian mixture of $ N(-1, \sigma^2)$ and $N(2, \sigma^2)$, with equal mixing weight.
> In this case, $\sigma$ is set as 0.3.
> We focused on the estimation performance for x in [-2, 2] and calculated the average absolute errors between the predicted confidence and the true confidence for the low region [0, 0.05] and the high region [0.95, 1].
> The results are shown in the following table.
>
> |    region   | square loss | cross entropy |
> | :----:| :----: | :----:|
> | [0,0.05] | 0.114 | 0.151 |
> | [0.95,1] | 0.113 | 0.149 |
>
> As can be seen, square loss calibrates better than cross entropy in the low and high regions.
>
>  As for the ECE plot on CIFAR dataset, the left pink bin (with no overlapping blue bin) is an artifact of the plotting program and should not have been plotted.
>  We hope to further clarify the meaning of Figure 2.
>  For CIFAR dataset, we cannot evaluate the calibration error as in Theorem 3.4, since we do not know the ground truth conditional probability.
>  In such case, we resort to the expected calibration error plot, where the x-axis in the ECE plot represents the predicted confidence of the model, and the y-axis is the test accuracy of samples whose predicted confidence falls in that range.
>
>  The message we want to send through the CIFAR-10 ECE experiment is that SL tends to be better calibrated.
>  However, the ECE plot has limitations and the results in Figure 2 should not be over-interpreted, due to the unknown ground truth conditional probability.
>  For a more rigorous verification, we refer the reviewer to the added simulated data experiment above, where we can evaluate $\hat{\eta}-\eta$ directly.

---

> > ### Author Response · Authors · 2022-08-02
> > **Response to Reviewer zn8W (Part 2)**
> >
> > ## Response to Q4
> >
> > Thank you for pointing it out. We mainly study the SL from the theoretical perspective in this work. We show that SL is theoretically sound for classification. However, we do not claim that SL is better than CE in all scenarios. Currently, we cannot rigorously analyze under what situation CE loss will outperform SL.
> >
> >  One important implication of our analysis is that
> >  SL would potentially be a good alternative loss function in various classification tasks, and it is worth to try SL in practice (like we may try different optimization methods in training a neural network). In Table 1, we can see that the neural network structures can also influence accuracy.
> >
> > Empirically, [Hui \& Belkin ICLR 2020] showed that SL tends to perform better for NLP tasks, but CE is generally better for CV tasks. In contrast to the empirical evaluations of SL,  theoretical understanding of SL in classification is still limited.
> > This work is the first to establish excess risk convergence rate and calibration error convergence rate for neural networks trained with square loss.
> >
> > ## Response to Limitations
> > This work mainly contributes to the theoretical understanding of SL in deep classification, where previously, no theoretical guarantee has been established.
> > We believe that there should be a lot of interesting research topics along this line, including the disadvantages of SL and how to improve it, which will be pursued in the future.
> >
> > In practice, the performance of a certain loss function can be very complicated to analyze because it can depend on so many factors, including data structure, optimization method, neural network architecture, etc. As we have mentioned, [Hui \& Belkin ICLR 2020] showed that SL tends to perform better for NLP tasks, but CE is generally better for CV tasks. We also found another interesting phenomenon in practical training.
> >
> >
> > We found through our experiments that SL may demand larger network capacity to overfit the training data. For example, in our added Tiny-ImageNet experiment, while the test accuracies are comparable, ResNet-18 trained with the square loss after 200 epochs can only attain less than 96\% training accuracy while that for CE is around 99.9\%.
> > This may be the optimization problem, which calls for further studies.

---

### Official Review · Reviewer_rSfq · 2022-07-10

**Rating:** 6
**Confidence:** 4
**Soundness:** 3 good
**Presentation:** 3 good
**Contribution:** 3 good

**Summary:**

The authors theoretically investigate the square loss in classification for over-parametrized neural networks in NTK regime. They reveal properties regarding the generalization error, robustness and calibration error. Separable and non-separable cases are considered. In the non-separable case, they establish fast convergence rate  for misclassification rate and calibration error. While in separable case, the misclassification rate improves to be exponentially fast.

Further, they prove a margin lower bounded away from zero, and claim theoretical guarantees for robustness. They also conduct experiments with practical neural networks to support and extend the theoretical claims.

The main conclusion is that compared with the cross-entropy loss, the square loss has comparable generalization error while noticeable advantages in robustness and model calibration.

**Questions:**

1) It was widely believed that training with the square loss in classification suffer from the 'outliers' problem, and it was not robust.
How do your claim on robustness solve the concerns on the 'outlier'  issue?

2) Are the results of the square loss in Table 1 with simplex coding? As in large-class number case, previous works show that the original square loss with one-hot encoding may not work well, it seems in Table 1 the square loss results on CIFAR-100 are way better than the cross-entropy. How simplex encoding make a difference with the one-hot encoding?

**Limitations:**

This paper extends the study of the square loss in classification by providing systematic theoretical study, which is import for people to consider other possible alternatives as training objectives in classification.

The assumptions are a little bit strong which may not hold in practice, and the experiments are a little bit limited for practitioners to be convinced.

Overall, the paper indeed provides more theoretical understanding which is an important contribution to the study of square loss in classification.

**Strengths And Weaknesses:**

Strengths:

1) This paper extends the study of square loss in classification with more theoretical understanding, with systematical investigations on several import properties: generalization error, robustness and model calibration.

2) The main conclusion is well-supported by theoretical proofs and some empirical results.

3) The paper is well-written and easy to follow.

Weaknesses:

The experimental results is not very convincing as it mainly considers CIFAR-10 dataset.

---

> ### Author Response · Authors · 2022-08-02
> **Response to Reviewer rSfq**
>
> ## Response to Weaknesses
>
> This work mainly contributes to the theoretical understanding of using square loss for neural network-based classification. Numerical experiments are conducted to corroborate our theoretical results, rather than establishing new state-of-the-art benchmarks.
> Hence, our experiments only considered fully connected ReLU networks on simulated data and (Wide) ResNet on the CIFAR dataset.
> For more comprehensive numerical experiments on larger-scale datasets, we refer the readers to [Hui \& Belkin ICLR 2020] and [Kornblith et al. 2020], as stated in lines 35-40 in our submission.
>
> We also added experiments on Tiny-ImageNet, where the findings remain the same, i.e., comparable clean accuracy but better adversarial robustness and calibration. The results are summarized below:
> |  | Clean Acc | AutoAttack($l_\infty $  2/255) | ECE |
> | :--:| :----: | :----:|:----:|
> | CE | 62.9% | 0. | 0.081|
> | SL | 62.7% | 0.49| 0.059|
>
> The network is ResNet-18. The batch size and learning rate are 128 and 0.2 respectively. The classifier is trained by SGD with weight decay (0.0001).  We run 3 replications and the reported number is the average.
>
>
> ## Response to Q1
> In our context, the robustness mainly refers to the  "adversarial robustness", namely, the model's robustness against samples that are intentionally designed to fool the model, with certain perturbation strength (often measured by $L_\infty$ norm). Mathematically speaking, the adversarial robustness is measured by the loss
>
> $L_{{\rm adv},\epsilon}(f) $=$ E_{(X,Y)\sim P}$$\left(\max_{x \in B(X,\epsilon)}I_{{\rm sign}(f(x)) \neq Y}\right)$ where $B(X,\epsilon)$ is an $L_\infty$ ball with radius (i.e., strength) $\epsilon$.
>
>  From this equation, it can be seen that the adversarial robustness usually is tied to the separable case, and a classifier with max-margin is preferred. This is different from the ``outlier'' problem, which refers that a sample that is far away from its own class will strongly influence the performance of the model. Following your question, we revise the statement about the robustness in the Introduction.
>
>  As for the "outlier'' issue, we intuitively think that because the neural network classifiers are often trained with some regularization, this regularization helps the model to perform well against the ``outliers''. This is verified in synthetic simulation; See Appendix G.1: Non-separable case for more details, where an outlier can appear.
>
> Furthermore, following your comments, we added the ResNet-18 on CIFAR-10 experiment to investigate SL under label noise. If a sample is given a wrong label, it can be thought of as an outlier. In the experiment, we randomly flip 20\% to 50\% of the training labels and evaluated the test performance. The results are shown in the table below.
>
> |  Acc   | 20% | 30% | 40% | 50% |
> | :----:| :----: | :----:|:----:|:----:|
> | CE | 94.8% |93.7% |91.0%  |82.0%  |
> | SL | 94.7% |93.6%  |91.2%  |83.4% |
>
> As can be seen, the classification accuracy is comparable between CE and SL.
>
> ## Response to Q2
> Yes. The square loss results in Table 1 are with simplex coding.
>  The original one-hot label coding had been shown to be problematic when the number of classes is large. Specifically, [Hui \& Belkin ICLR 2020] proposed a rescaling mechanism for square loss and suggested that such modification is necessary when the number of classes is larger than 50.
>  In our work, we did not utilize such modifications to the one-hot coding and directly used the simplex coding. The comparison is discussed in the first paragraph of Section 3.4.
>  To reiterate, we do not think the difference between one-hot vs. simplex coding in SL is significant, certainly not as significant as SL vs. CE.
>
>  We believe their difference is more on the optimization level. In order to be more separated, the label coding should be as distant as possible. Vertices of a k-simplex correspond to maximally separated k points on a sphere. In contrast, one-hot coding only takes up the "positive" part of the feature space. In our experiments with CIFAR-100 and Tiny-ImageNet, the simplex coding seems to work fine.
>
> ## Response to Limitations
> Thank you for your comments. In the future, we will work on relaxing the assumptions of the theoretical results to make them closer to practical settings. For example, we intend to relax the two-layer neural network assumption to multi-layer neural networks.
> Although our work focuses on a theoretical perspective, the improved theoretical understanding does translate to practical settings, where we found that SL can be advantageous in robustness and calibration.
> We added the ResNet on Tiny-ImageNet experiment, where there are 200 classes and we intend to add more experiments in future works.

---

### Meta-Review · Area_Chair_uwrh · 2022-08-29

**Recommendation:** Accept
**Confidence:** Certain

**Metareview:**

This paper provides a theoretical investigation of square loss in the NTK regime due to a recent observation: empirical evidence seems to promote square loss other than cross-entropy loss. The authors provide generalization error (in this work it refers to population risk) bound, calibration error, and robustness (in terms of lower bounds of the margin). The theoretical analysis justifies the benefits of square loss. Reviewers all agree that the analysis is novel. Three reviewers think the experimental results might need further improvement. I suggest the authors to strengthen the experiments in the final version. Overall, I recommend accept.


**Award:**

No

---

### Decision · Program_Chairs · 2022-09-14

Accept